# Towards Fast, Specialized Machine Learning Force Fields: Distilling Foundation Models via Energy Hessians

**Ishan Amin,**[*] **Sanjeev Raja**[*]**, and Aditi S. Krishnapriyan**
University of California, Berkeley; LBNL
{ishanthewizard,sanjeevr,aditik1}@berkeley.edu

## ABSTRACT

The foundation model (FM) paradigm is transforming Machine Learning Force Fields (MLFFs), leveraging general-purpose representations and scalable training to perform a variety of computational chemistry tasks. Although MLFF FMs have begun to close the accuracy gap relative to first-principles methods, there is still a strong need for faster inference speed. Additionally, while research is increasingly focused on general-purpose models which transfer across chemical space, practitioners typically only study a small subset of systems at a given time. At test time, MLFFs must also obey physical constraints unique to the downstream use case, such as energy conservation for molecular dynamics simulations. This underscores the need for fast, specialized MLFFs relevant to specific downstream applications, which preserve test-time physical soundness while maintaining train-time scalability. In this work, we introduce a method for transferring general-purpose representations from MLFF foundation models to smaller, faster MLFFs specialized to specific regions of chemical space. We formulate our approach as an architecture-agnostic knowledge distillation procedure, where the smaller "student" MLFF is trained to match the Hessians of the energy predictions of the "teacher" foundation model. We demonstrate our approach across multiple recent foundation models, large-scale datasets, chemical subsets, and downstream tasks. Our specialized MLFFs can be up to $20 \times$ faster than the original foundation model, while retaining, and in some cases exceeding, its performance and that of undistilled models. We also show that distilling from a teacher model with a direct force parameterization into a student model trained with conservative forces (i.e., computed as derivatives of the potential energy) successfully leverages the representations from the large-scale teacher for improved accuracy, while maintaining energy conservation during test-time molecular dynamics simulations. More broadly, our work suggests a new paradigm for MLFF development, in which foundation models are released along with smaller, specialized simulation "engines" for common chemical subsets. The implementation of our method is available at https://github.com/ASK-Berkeley/MLFF-distill.

## 1 INTRODUCTION

Quantum chemical calculations, such as Density Functional Theory (DFT), underpin a broad range of applications in computational chemistry, including the discovery of new drugs (Cole & Hine, 2016), materials (Hafner et al., 2006; Jain et al., 2016), and catalysts (Hammer & Nørskov, 2000). Machine learning force fields (MLFFs) (Gasteiger et al., 2021; Batzner et al., 2022; Musaelian et al., 2022; Batatia et al., 2022) based on graph neural network (GNN) architectures (Gilmer et al., 2017) have recently shown tremendous potential to serve as fast surrogates for these quantum mechanical calculations.

Foundation models (FMs) are general-purpose models trained on large quantities of data, with the ability to generalize to many downstream tasks with little to no fine-tuning. Mirroring advancements in the fields of natural language processing (Achiam et al., 2023) and computer vision (Radford et al., 2021; Oquab et al., 2023), the availability of increasingly large and diverse datasets of quantum chemical calculations (Chanussot et al., 2021; Jain et al., 2020; Eastman et al., 2023) has enabled the creation of MLFF FMs (Kovács et al., 2023; Batatia et al., 2023; Shoghi et al., 2023). While earlier MLFFs typically trained on relatively narrow datasets (Schütt et al., 2018; Chmiela et al., 2017), MLFF FMs are trained across a broad swath of chemical space, aiming to perform well across a diverse range of atomic property prediction tasks.

---

[*]Equal Contribution.

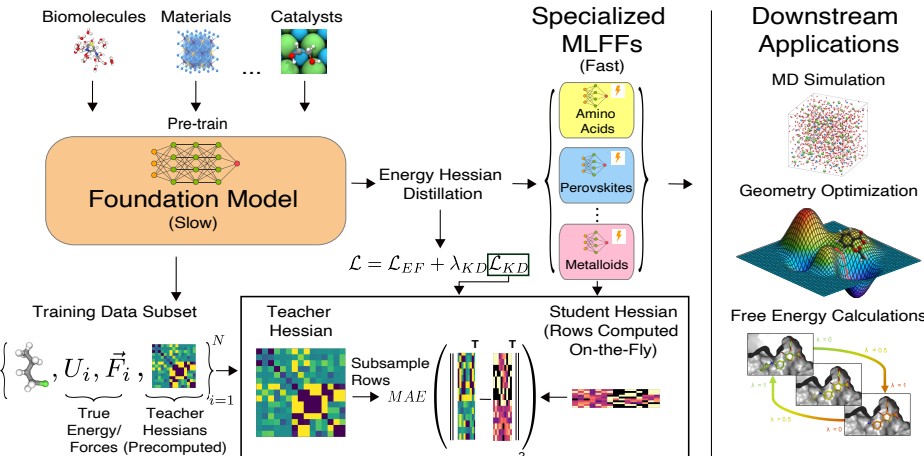

Figure 1: **Proposed Hessian distillation schematic.** In our proposed distillation approach, we start with a machine learning force field (MLFF) foundation model (FM) that has been trained on a large quantity of diverse data. We precompute energy Hessians of the FM over a specialized data subset. We then train a series of smaller MLFFs on these subsets via our knowledge distillation loss ($\mathcal{L}_{KD}$), which aligns selected rows of the energy Hessian of the smaller (student) models with those of the FM (teacher). We also keep the conventional procedure of training on the ground truth energies and forces ($\mathcal{L}_{EF}$) from the specialized subset. The resulting MLFFs are considerably faster than the FM and can be efficiently used in downstream applications such as MD simulation, geometry optimization, and free energy calculations.

While MLFF FMs trained on large quantities of *ab-initio* data have begun to approach the accuracy of DFT for some tasks, there are still significant challenges in improving efficiency for modeling large time and length scales. In line with the increasing size and diversity of training data, MLFFs have steadily grown in complexity, both in terms of raw parameter count and design choices such as the use of expensive tensor products to enforce higher-order Euclidean symmetries (Batzner et al., 2022; Batatia et al., 2022). Despite efficiency efforts (Luo et al., 2024), state-of-the-art MLFFs remain several orders of magnitude slower than alternatives such as classical force fields (Unke et al., 2021; Wang et al., 2024). As a result, MLFF FMs are often prohibitively expensive to use in realistic downstream applications, such as molecular dynamics (MD) simulations with $> 10^6$ timesteps.

Furthermore, a clear tension arises between the need for fast, scalable training of MLFFs, and ensuring physical consistency when the model is deployed in downstream applications such as MD simulations. For example, trying to build constraints into the architecture such as SO(3) equivariance can slow down training. Restricting training to only an equivariant space may also complicate optimization and training dynamics, consequently affecting the quality of the final model. Although there is increasing evidence that SO(3) equivariance can be learned accurately directly from data (Qu & Krishnapriyan, 2024), other constraints, such as energy conservation, are more difficult to enforce without hard constraints (Bigi et al., 2024), but are critical in downstream applications such as MD simulations.

More broadly, the increasing generality of MLFF FMs is at odds with the needs of practitioners, who are often ultimately focused on a relatively narrow set of systems and downstream applications (perovskites, magnesium-based electrolytes, insulators, etc.). Fine-tuning the FM for these downstream applications is in principle straightforward, but may be prohibitive for many practitioners and offers no speedup at inference time. These challenges motivate us to ask the following questions:

1. **How can we improve the efficiency of MLFFs for specialized tasks while preserving the powerful general-purpose representations learned by FMs?**

2. **How can we ensure that test-time physical soundness, such as energy conservation for molecular dynamics simulations, is maintained while preserving train-time scalability?**

To address this, we introduce an approach based on knowledge distillation (KD) which learns fast, specialized MLFFs from large, general-purpose FMs. The core of our approach is a training objective that aligns the Hessians of the energy predictions between the foundation MLFF (teacher) and specialized MLFF (student). The method is conceptually simple and efficient: the Hessians of the FM can

be pre-computed once and stored, while the student Hessian computation can be accelerated using approximate sampling techniques. Unlike existing KD methods which align internal features between the teacher and student (Kelvinius et al., 2023), our approach is entirely agnostic to model architecture and inductive biases, and can be used out-of-the-box for any student-teacher pairing. We demonstrate our approach on four MLFF FMs: MACE-OFF (Kovács et al., 2023) and EScAIP (Qu & Krishnapriyan, 2024) trained on SPICE (Eastman et al., 2023), MACE-MP-0 (Batatia et al., 2023) trained on MPtrj (Deng et al., 2023) from the Materials Project (Jain et al., 2020), and JMP (Shoghi et al., 2023) finetuned on selected molecules from MD22 (Chmiela et al., 2023). We learn student MLFFs specialized to subsets of the FM's training distribution which mimic realistic downstream applications, such as specifically modeling amino acids or materials containing Yttrium. These specialized student models achieve inference speeds up to 20 times faster than the original FMs. Our approach also achieves substantial improvements in energy and force error, MD simulation stability, energy conservation, and geometry optimization, compared to student models trained without distillation. In most cases, the student models also outperform the original FM. To our knowledge, this is the first approach to create fast, specialized MLFFs from FMs.

## 2 BACKGROUND AND RELATED WORK

**Machine Learning Force Fields.** A Machine Learning Force Field (MLFF) is a learnable function approximator $U_\theta$ which maps a molecular configuration to a potential energy and per-atom forces. Specifically, it takes the positions of $n$ atoms, $\mathbf{r} = (\mathbf{r}^{(1)}, ..., \mathbf{r}^{(n)}) \in \mathbb{R}^{n \times 3}$, and their atomic numbers, $\mathbf{z} = (z^{(1)}, ..., z^{(n)}) \in \mathbb{R}^n$ as inputs, and outputs a potential energy $U_\theta \in \mathbb{R}$ and per-atom forces $\mathbf{F}_\theta = (\mathbf{f}_\theta^{(1)}, ..., \mathbf{f}_\theta^{(n)}) \in \mathbb{R}^{n \times 3}$. MLFFs are typically parameterized as graph neural networks (GNNs), and are trained via the following regression loss, with supervision from a dataset of reference energies and forces:

$$\mathcal{L}_{EF} = \lambda_U |U_{\text{ref}}(\mathbf{z}, \mathbf{r}) - U_\theta(\mathbf{z}, \mathbf{r})|^2 + \lambda_F \sum_{i=1}^{n} \|\mathbf{f}_{\text{ref}}^{(i)}(\mathbf{z}, \mathbf{r}) - \mathbf{f}_\theta^{(i)}(\mathbf{z}, \mathbf{r})\|_2^2. \tag{1}$$

**MLFF Inductive Biases.** MLFF speed, accuracy, physical soundness, and ease of training are affected by the choice of inductive biases. SO(3) equivariant networks (Geiger & Smidt, 2022) guarantee that the forces, and possibly internal features, rotate consistently when the positions of the input rotate, but can be both hard to optimize over and considerably slower than non-equivariant models, often due to the reliance on expensive tensor products to handle SO(3) and spherical harmonic representations. Another key inductive bias is a conservative force parameterization. MLFFs which obtain conservative forces by differentiating the energy output with respect to the atomic coordinates ($\mathbf{F}_\theta = -\nabla_\mathbf{r} U_\theta$) guarantee conservation of the model's energy in MD simulations, while MLFFs parameterizing the force separately from the energy lack this guarantee but are faster. There is growing evidence that SO(3) equivariance can be learned without explicit enforcement, given sufficient data and model scaling and/or data augmentation (Neumann et al., 2024; Qu & Krishnapriyan, 2024; Pozdnyakov & Ceriotti, 2024), while energy conservation is harder to learn without conservative forces ((Bigi et al., 2024)). In §4, we demonstrate that our distillation approach works well with many combinations of teacher and student model inductive biases.

**MLFF Foundation Models.** While early MLFFs were trained on relatively narrow datasets, foundation models (FMs) trained across a diverse swath of chemical space are now becoming increasingly common (Shoghi et al., 2023; Gasteiger et al., 2022; Batatia et al., 2023; Kovács et al., 2023). MACE-OFF (Kovács et al., 2023) was primarily trained on a filtered subset of 951,000 biomolecular structures from the SPICE (Eastman et al., 2023) dataset. MACE-MP0 (Batatia et al., 2023) was trained on 1.6 million structures from the Materials Project (Jain et al., 2020). The JMP (Shoghi et al., 2023) FM was pre-trained on a combined dataset consisting of OC20, OC22, ANI-1x, and Transition-1x, and later fine-tuned on several datasets such as QM9, rMD17, and MD22. The promise of MLFFs lies in their ability to be used zero-shot or with minimal finetuning across many downstream tasks. However, as MLFF FMs have increased in complexity and scale to match the diversity and size of training data, speed has become a limiting factor, particularly in modeling systems with large time and length scales (Unke et al., 2021; Wang et al., 2024).

**Knowledge Distillation.** Knowledge distillation (KD) (Hinton et al., 2015) aims to transfer knowledge from a larger teacher model to a smaller student model, usually by training the student to mimic certain properties of the teacher (Romero et al., 2015; Sanh et al., 2019; Tang et al., 2020; Gou et al., 2021). This is typically done by minimizing a distillation objective of the form $L_{\text{KD}} = \mathbb{E}_x \|\phi_T(x) - P\phi_S(x)\|_2^2$, where $\phi_T$ and $\phi_S$ are intermediate features of the teacher and student respectively, and $P$ is a linear projection that accounts for differences in dimensionality between the two models. Specializing FMs to specific subdomains

has been explored in large-scale language and vision models (Qiu et al., 2024), but, to our knowledge, is unexplored in the context of MLFFs. Previous work on KD for MLFFs was done in (Kelvinius et al., 2023) by aligning node and edge features across models such as GemNet-OC, PaiNN, and SchNet on the OC-20 and COLL datasets. However, the best-performing method, referred to as "node-to-node" (**n2n**), did not specialize the models, and evaluated the student and teacher models on the same data. We demonstrate in §4 that our Hessian distillation approach consistently outperforms **n2n** across several datasets and MLFFs.

**Learning from Function Derivatives.** Sobolev training (Czarnecki et al., 2017) uses function derivatives as supervision to train neural networks, including for KD. Their work highlights numerous theoretical benefits of training to match function derivatives, including better sample complexity and reduced overfitting. To our knowledge, this form of training has not been used to specialize to a subset of the data distribution.

## 3 DISTILLING FOUNDATION MODELS WITH ENERGY HESSIANS

After reviewing background on energy Hessians in §3.1, we introduce our method for producing fast, specialized MLFFs via knowledge distillation (KD) from the energy Hessians of pre-trained foundation models in §3.2. In our setting, the FM plays the role of the teacher, while the fast, specialized MLFF is the student. We also present our Hessian subsampling strategy to significantly accelerate training (§3.3).

### 3.1 BACKGROUND ON ENERGY HESSIANS

The Hessian of the energy is the second derivative of the energy with respect to atomic positions, or equivalently, the negative derivative of the forces with respect to the positions. Given $3N$-dimensional unrolled force and position vectors, the Hessian $\mathbf{H} \in \mathbb{R}^{3N \times 3N}$ is given by $\mathbf{H} = -\frac{\partial \mathbf{F}}{\partial \mathbf{r}} = \frac{\partial^2 U}{\partial \mathbf{r}^2}$. Accordingly, $\mathbf{H}_{ij}$ is the derivative of the $i^{th}$ force with respect to the $j^{th}$ position.

### 3.2 ENERGY HESSIAN ALIGNMENT OBJECTIVE

Given a dataset of $N$ molecular structures paired with quantum-mechanical energy and force labels $\mathcal{D} = \{(\mathbf{z}_i, \mathbf{r}_i, U_i, \mathbf{F}_i)\}_{i=1}^N$, and a MLFF FM $T_\psi$ pretrained on $\mathcal{D}$ that predicts energies $U_\psi$ and forces $F_\psi$, we first precompute the Hessians of the FM energy predictions over the dataset $\mathcal{D}$ using automatic differentiation (we demonstrate that finite differences can also be used in §A.11). This results in an augmented dataset $\mathcal{D}_{aug} = \{(\mathbf{z}_i, \mathbf{r}_i, U_i, \mathbf{F}_i, \mathbf{H}_i)\}_{i=1}^N$, where $\mathbf{H}_i = \frac{\partial^2 U_\psi(\mathbf{z}_i, \mathbf{r}_i)}{\partial \mathbf{r}^2}$. We then train a student MLFF $S_\phi$, assumed to be small relative to $T_\psi$ (i.e., $|\phi| << |\psi|$), on a subset of the data, $\mathcal{D}_{KD} \subset \mathcal{D}_{aug}$. This subset corresponds to a specific downstream application, such as molecules with Iodine. Crucially, in addition to matching the energies and forces of $\mathcal{D}_{KD}$, we also train the student $S_\phi$ to match the FM Hessians over $\mathcal{D}_{KD}$.

The complete loss function for training the student via knowledge distillation over the subset $\mathcal{D}_{KD}$ is:

$$\mathcal{L}(\phi) = \mathbb{E}_{\mathbf{z}_i, \mathbf{r}_i, \mathbf{H}_i \sim \mathcal{D}_{KD}} \left[ \mathcal{L}_{EF}(\phi) + \lambda_{KD} \| \mathbf{H}_i + \frac{\partial F_\phi(\mathbf{z}_i, \mathbf{r}_i)}{\partial \mathbf{r}} \|_2^2 \right], \tag{2}$$

where $\mathcal{L}_{EF}$ is the standard energy and force-matching objective defined in Eq. 1 and $\lambda_{KD}$ is a hyperparameter controlling the strength of KD. We highlight that for direct-force student MLFFs, the energy Hessian is computed as the negative Jacobian of the force, rather than the second derivative of the energy prediction.

In practice, we find that the student can outperform the FM on the original objective ($\mathcal{L}_{EF}$) due to the reduced diversity of $\mathcal{D}_{KD}$ and the regularization effect of teacher supervision. To ensure that the student performance is not bottlenecked by the FM, we reduce the weight of the Hessian distillation loss term, $\lambda_{KD}$, by a factor of 2 during training once the student's validation loss on the original objective, $\mathcal{L}_{EF}(\phi)$, becomes lower than that of the frozen FM, $\mathcal{L}_{EF}(\psi)$ (see §A.12 for more details).

### 3.3 IMPROVING EFFICIENCY OF HESSIAN COMPUTATIONS WITH SUBSAMPLING

Obtaining the energy Hessian for a molecule via autodifferentiation requires $3N$ backwards passes, one per force value. To mitigate this computational expense, we instead uniformly sample rows from the reference Hessian on which to supervise in each training iteration. We accordingly only compute these rows of the student's Hessian. Each row corresponds to one Euclidean coordinate of a single atom. Formally, let $\mathcal{J}_i \subset \{1, ..., 3N\}$ be the set of $s$ randomly sampled indices corresponding to the rows of the Hessian for a particular molecular structure: $\mathcal{J}_i = [j_1, \cdots j_s]$. For each sample in the dataset, we supervise the student model on the subset $\mathcal{J}_i$ of Hessian rows. The modified loss function becomes,

$$\mathcal{L}(\phi) = \mathbb{E}_{\mathbf{z}_i, \mathbf{r}_i, \mathbf{H}_i \sim \mathcal{D}_{KD}} \left[ \mathcal{L}_{EF}(\phi) + \lambda_{KD} \cdot \mathbb{E}_{\mathcal{J}_i \sim \mathcal{U}_s(1, 3N)} \left( \frac{1}{s} \sum_{j \in \mathcal{J}_i} \left\| \mathbf{H}_i^{(j)} + \frac{\partial F_\phi^{(j)}(\mathbf{z}_i, \mathbf{r}_i)}{\partial \mathbf{r}} \right\|_2^2 \right) \right], \tag{3}$$

where $\mathcal{U}_s(1,3N)$ denotes the uniform distribution over subsets of $s$ rows from the Hessian, and $\mathbf{H}_i^{(j)}$ and $F_\phi^{(j)}$ are the $j$-th row of the reference Hessian and student forces, respectively. The number of backward passes required to compute the Hessian grows as $\mathcal{O}(s)$, so subsampling significantly accelerates training. Reducing the number of sampled rows to $s=1$ does not noticeably impact model performance (§5).

**Computing Hessian Rows via Vector-Jacobian Products.** When computing individual rows, we wish to avoid forming the entire Hessian matrix. We achieve this by using vector-Jacobian products (VJPs), the fundamental operation underlying reverse-mode autodifferentiation. Formally, given a function $f(\mathbf{x}):\mathbb{R}^{d_{in}} \to \mathbb{R}^{d_{out}}$ and a vector $\mathbf{v} \in \mathbb{R}^{d_{out}}$, reverse-mode autodifferentiation computes the VJP $\mathbf{v}^\top \mathbf{J}$ in a matrix-free manner (e.g., without explicitly forming the Jacobian $\mathbf{J}=\partial_{\mathbf{x}}f \in \mathbb{R}^{d_{out} \times d_{in}}$). In our setting, $f$ is the MLFF force $\mathbf{F}:\mathbb{R}^{3N} \to \mathbb{R}^{3N}$, whose Jacobian is the energy Hessian $\mathbf{H} \in \mathbb{R}^{3N \times 3N}$. To extract the $j^{th}$ Hessian row, we construct a one-hot vector $\mathbf{v}=\mathbf{e}_j \in \mathbb{R}^{3N}$ to use in the VJP. Let $\mathbf{P}_{\mathcal{J}} \in \mathbb{R}^{s \times 3N}$ denote the permutation matrix containing one-hot vectors corresponding to the subset of sampled row indices in $\mathcal{J}$: $\mathbf{P}_{\mathcal{J}} = [\mathbf{e}_{j_1},\cdots,\mathbf{e}_{j_s}]^\top$. By computing the matrix-Hessian product $\mathbf{P}_{\mathcal{J}}\mathbf{H}$, which is achieved via a `vmap` over the rows of $\mathbf{P}_{\mathcal{J}}$ of the force VJP, we can efficiently extract the desired Hessian rows $(\mathbf{H}^{(j_1)},\mathbf{H}^{(j_2)},...,\mathbf{H}^{(j_s)})^\top$.

### 3.4 Energy Gradient Supervision for Direct Force Models

To improve the energy predictions of MLFFs with a direct force parameterization, we find it useful to indirectly utilize the inductive bias that the force is the negative gradient of the energy. Specifically, we introduce an additional loss term to align the negative gradient of the energy head with the true forces: $\mathcal{L}_{\nabla U}=\|\mathbf{F}+\nabla_{\mathbf{r}}U_\theta\|^2$, with loss weight $\lambda_{\nabla U}$. Here, $\mathbf{F}$ represents the true forces, and $\nabla_{\mathbf{r}}U_\theta$ is the gradient of the predicted energy. Importantly, no gradients of the energy head are computed at inference time (forces are derived as usual through the separate force head), so there is no impact on inference speed. See §A.13 for more details.

## 4 Experimental Results

We present the results of our Hessian distillation approach for learning fast, specialized MLFFs from large foundation models. In §4.1, we distill the MACE-OFF FM, which was trained on the SPICE biomolecules dataset. We also distill an EScAIP Qu & Krishnapriyan (2024) FM trained on SPICE in §A.1. In §4.2, we distill the MACE-MP0 model trained on the MPtrj dataset. Finally, in §4.3, we distill the JMP FM, which was pretrained on several large datasets and finetuned on selected MD22 molecules. In each setting, we train fast, specialized MLFF students on data subsets corresponding to realistic downstream applications.

We compare against the following baselines for training student MLFFs (more details in §A.6):

1. **Undistilled**: Training on the specialized data subset without Hessian supervision (i.e., $\lambda_{KD}=0$).
2. **n2n**: Training to match the node representations of the FM at the final layer. This is a direct comparison to the best-performing MLFF KD technique introduced in (Kelvinius et al., 2023).
3. **a2a**: Training on top of a learned projection of the FM's atom embeddings. We create this baseline in the spirit of self-supervised learning techniques which fine-tune the representations of a large model for specific downstream tasks (Devlin, 2018; Radford et al., 2021; Caron et al., 2021).

We measure MD simulation speed by performing inference with a batch size of 64 on a single NVIDIA A6000 GPU and converting it to nanoseconds per day by assuming a 1 femtosecond timestep. This mimics real-world use cases of performing parallel MD simulations, vectorized over the batch dimension, to rapidly explore molecular phase space, or for high-throughout structure screening. We also measure speed using a batch size which maximizes sample throughput for each model. These batch sizes are given in §A.7. We note that in production MD simulations, MLFFs would be linked with an optimized platform like LAMMPs (Thompson et al., 2022), potentially leading to significant acceleration beyond the numbers we report.

### 4.1 Distilling MACE-OFF on SPICE

Using MACE-OFF (Kovács et al., 2023) as the teacher model, we focus on small subsets of SPICE with limited amounts of data—monomers, solvated amino acids, and molecules containing iodine—to train small, specialized GemNet-dT, PaiNN, and GemNet-T student MLFFs via our Hessian distillation approach described in §3.2. While the FM and specialized student MLFFs are trained on different quantities of data, we report force mean absolute error (MAE) on test data not seen by either model during training. Results are shown in Tab. 1, with additional details and hyperparameter sweeps in §A.5 and §A.8.

Table 1: Results of distilling the MACE-OFF foundation model trained on SPICE into specialized MLFFs. (FM) indicates foundation model, while (S) indicates student model. **n2n** is the node feature matching baseline from (Kelvinius et al., 2023), while **a2a** is the atom embedding matching baseline we construct in §A.6. Student models all have identical simulation speeds. Speedups relative to FM are given in parentheses. The "Speed" column is calculated with a constant batch size of 64 for all models, while the "Maximum Speed" is calculated with the batch size that maximizes throughput for each model.

| Chemical Subgroup | Dataset Size | Model (Parameter Count) | Distillation Method | Force MAE (meV/Å) (↓) | Energy MAE (meV/atom) (↓) | Speed (ns/day) (↑) | Maximum Speed (ns/day) (↑) |
|---|---|---|---|---|---|---|---|
| Monomers | 14,331 | (FM) MACE-OFF Large (4.7M) | – | 6.6 | 0.65 | 38.0 | 38.1 |
| | | (S) GemNet-dT (0.67M) | Undistilled | 11.3 | 1.27 | | |
| | | | n2n | 10.5 | 1.2 | | |
| | | | a2a | 12.9 | 1.6 | | |
| | | | Hessian (ours) | **6.3** | **0.4** | **164.5** (4.3x) | **725.2** (19.0x) |
| | | (S) PaiNN (1.0M) | Undistilled | 25.0 | 2.3 | | |
| | | | n2n | 20.8 | 1.5 | | |
| | | | a2a | 24.7 | 2.3 | | |
| | | | Hessian (ours) | **8.77** | **0.48** | **291.5** (7.7x) | **1827** (48.0x) |
| | | (S) GemNet-T (0.67M) | Undistilled | 7.9 | **3.8** | | |
| | | | n2n | 7.7 | 5.3 | | |
| | | | a2a | 7.2 | 4.0 | | |
| | | | Hessian (ours) | **5.1** | 6.8 | **66.0** (1.74x) | **408.9** (10.8x) |
| Solvated Amino Acids | 805 | (FM) MACE-OFF Large (4.7M) | – | 19.4 | 1.3 | 3.8 | 3.8 |
| | | (S) GemNet-dT (0.67M) | Undistilled | 22.4 | 2.2 | | |
| | | | n2n | 20.7 | 1.6 | | |
| | | | a2a | 24.4 | 1.6 | | |
| | | | Hessian (ours) | **11.6** | **0.37** | **44.4** (11.7x) | **44.4** (11.7x) |
| | | (S) PaiNN (1.0M) | Undistilled | 50.1 | 3.3 | | |
| | | | n2n | 38.3 | 1.7 | | |
| | | | a2a | 52.4 | 3.7 | | |
| | | | Hessian (ours) | **18.0** | **0.41** | **79.4** (20.9x) | **79.4** (20.9x) |
| | | (S) GemNet-T (0.67M) | Undistilled | 18.3 | 1.7 | | |
| | | | n2n | 18.0 | 1.8 | | |
| | | | a2a | 17.1 | 1.9 | | |
| | | | Hessian (ours) | **11.2** | **1.2** | **23.4** (6.2x) | **23.4** (6.2x) |
| Systems with Iodine | 11,171 | (FM) MACE-OFF Large (4.7M) | – | 15.3 | 1.3 | 14.8 | 14.8 |
| | | (S) GemNet-dT (0.67M) | Undistilled | 23.4 | 2.68 | | |
| | | | n2n | 23.3 | 2.3 | | |
| | | | a2a | 23.2 | 2.6 | | |
| | | | Hessian (ours) | **14.7** | **0.58** | **148.1** (10.0x) | **220.4** (14.9x) |
| | | (S) PaiNN (1.0M) | Undistilled | 51.2 | 3.3 | | |
| | | | n2n | 43.6 | 2.3 | | |
| | | | a2a | 50.7 | 3.5 | | |
| | | | Hessian (ours) | **23.7** | **0.88** | **270.2** (18.3x) | **440.7** (29.8x) |
| | | (S) GemNet-T (0.67M) | Undistilled | 15.9 | **4.0** | | |
| | | | n2n | 15.9 | 4.4 | | |
| | | | a2a | 15.9 | 5.6 | | |
| | | | Hessian (ours) | **11.7** | 5.8 | **65.4** (4.4x) | **128.0** (8.6x) |

Using the distilled, specialized MLFFs, we achieve up to $20\times$ increases in simulation speed relative to the FM, and up to $50\times$ increases for throughput-maximizing batch sizes. For all splits, our Hessian distillation approach significantly outperforms training without distillation, as well as the **a2a** and **n2n** baselines, on Energy and Force MAE. In many cases, our distilled models outperform the FM, likely because they can focus all of their expressivity towards learning a narrower slice of chemical space. We report the times required to train the distilled student models relative to that of the original FM in §A.5. These times are generally nominal; when sampling 4 rows of the Hessian ($s = 4$), training the distilled student model requires an average of 4.0% additional compute beyond FM training. We demonstrate the downstream usefulness of our distilled models in constant-temperature MD simulations in §A.14. Our results indicate that our distilled student models are more stable than their undistilled counterparts. We also perform geometry optimization in §A.15, and find that our distilled models generally converge to structures with lower energy and force norms. We note that distilling a GemNet-T student model occasionally worsens energy accuracy relative to an undistilled model, even though the forces derived from these energies remain unaffected. This would not practically impact MD simulations, for which only forces are needed; however, if energies are important,

Table 2: Results of distilling the MACE-MP0 foundation model trained on MPtrj into specialized MLFFs. (FM) indicates foundation model, while (S) indicates student model. **n2n** is the node feature matching baseline from (Kelvinius et al., 2023), while **a2a** is the atom embedding matching baseline we construct in §A.6. Student models all have identical simulation speeds. Speedups relative to FM are given in parentheses. The "Speed" column is calculated with a constant batch size of 64 for all models, while the "Maximum Speed" is calculated with the batch size that maximizes throughput for each model.

| Chemical Subgroup | Dataset Size | Model (Parameter Count) | Distillation Method | Force MAE (meV/Å) (↓) | Speed (ns/day) (↑) | Maximum Speed (ns/day) (↑) |
|---|---|---|---|---|---|---|
| $Pm\bar{3}m$ Spacegroup | 9,725 | (FM) MACE-MP0 (15.8 M) | – | 18.1 | 93.6 | 101.9 |
| | | (S) GemNet-dT (0.67M) | Undistilled | 15.7 | | |
| | | | n2n | 14.6 | | |
| | | | a2a | 16.9 | | |
| | | | Hessian (ours) | **11.8** | **162.7** (1.7x) | **260.1** (2.6x) |
| | | (S) PaiNN (1.0M) | Undistilled | 21.9 | | |
| | | | n2n | 19.5 | | |
| | | | a2a | 23.3 | | |
| | | | Hessian (ours) | **15.5** | **264.4** (2.8x) | **451.5** (4.4x) |
| Systems with Yttrium | 30,436 | (FM) MACE-MP0 (15.8M) | – | 45.2 | 26.5 | 27 |
| | | (S) GemNet-dT (0.67M) | Undistilled | 32.5 | | |
| | | | n2n | 36.5 | | |
| | | | a2a | 36.5 | | |
| | | | Hessian (ours) | **21.3** | **73** (2.8x) | **73.3** (2.7x) |
| | | (S) PaiNN (1.0M) | Undistilled | 55.5 | | |
| | | | n2n | 37.7 | | |
| | | | a2a | 49.8 | | |
| | | | Hessian (ours) | **25.7** | **215.5** (8.1x) | **267.2** (9.9x) |
| Band Gap $\geq$ 5 meV | 36,150 | (FM) MACE-MP0 (15.8 M) | – | 31.4 | 13.4 | 13.4 |
| | | (S) GemNet-dT (0.67M) | Undistilled | 17.1 | | |
| | | | n2n | 15.1 | | |
| | | | a2a | 16.3 | | |
| | | | Hessian (ours) | **12.1** | **38.7** (2.9x) | **38.7** (2.9x) |
| | | (S) PaiNN (1.0M) | Undistilled | 32.6 | | |
| | | | n2n | 27.5 | | |
| | | | a2a | 32.2 | | |
| | | | Hessian (ours) | **16.3** | **125.4** (9.4x) | **125.4** (9.4x) |

coefficient rescaling to upweight the energy loss term, or energy fine-tuning as seen in (Kovács et al., 2023), would likely alleviate this issue. In §A.1, we demonstrate distillation with an EScAIP Qu & Krishnapriyan (2024) FM on SPICE, which leads to improved student model performance and eliminates this energy issue.

## 4.2 DISTILLING MACE-MP-0 ON MATERIALS PROJECT

We next consider MACE-MP-0 (Batatia et al., 2023), trained on 1.6 million structures from the MPtrj (Deng et al., 2023) dataset, as a teacher model, choosing the following subsets on which to learn specialized Gemnet-dT and PaiNN student MLFFs: materials in the $Pm\bar{3}m$ spacegroup (which includes cubic perovskites used in photovoltaic devices), materials containing Yttrium (used in lasers and alloys), and materials with a band gap of greater than 5 meV (roughly corresponding to insulators). While DFT with the PBE functional is known to underestimate band gap (Mori-Sánchez et al., 2008), we assume this delineation is sufficient for our purposes of creating broad chemical subgroups. The results are shown in Tab. 2.

We find that the specialized student MLFFs obtained via our Hessian KD approach are up to $10\times$ faster than the original FM, and consistently outperform the undistilled, n2n, and a2a baselines in Force MAE across all splits. Interestingly, we find that the GemNet-dT student models outperform the FM even before distillation, and Hessian KD subsequently further improves the student models. We speculate that in this scenario, Hessian KD has a regularizing effect which enables the student to learn better representations despite distilling from a teacher with higher Force MAE, analogous to training with soft or noisy labels (Szegedy et al., 2016; Müller et al., 2019). We also note that the n2n and a2a methods both generally improve over the undistilled baseline, unlike with MACE-OFF on SPICE. This suggests that improvements from KD are not always correlated to the accuracy of the teacher model.

Table 3: Results of distilling JMP-Large (JMP-L) and JMP-Small (JMP-S) foundation models finetuned on selected large MD22 molecules into specialized MLFFs. (FM) indicates foundation model, while (S) indicates student model. Student model speedups relative to the JMP-L FM are given in parentheses. The "Speed" column is calculated with a constant batch size of 1 for all models, while the "Maximum Speed" is calculated with the batch size that maximizes throughput for each model.

| Molecule | Dataset Size | Model (Parameter Count) | Distillation Method | Force MAE (meV/Å) (↓) | Energy MAE (meV/ atom) (↓) | Speed (ns/day) (↑) | Maximum Speed (ns/day) (↑) |
|---|---|---|---|---|---|---|---|
| Buckyball Catcher | 600 | (FM) JMP-S (39.9M) | – | 7.8 | – | 0.8 | 1.8 |
| | | (FM) JMP-L (220M) | – | 4.3 | – | 0.4 | 0.6 |
| | | (S) GemNet-dT (0.67M) (Direct-Forces, Invariant) | Undistilled | 8.0 | 1.0 | | |
| | | | JMP-S Hessian (ours) | **5.1** | **0.15** | | |
| | | | JMP-L Hessian (ours) | **5.1** | **0.15** | **2.4** (6x) | **18.3** (30.5x) |
| | | (S) GemNet-T (0.57M) (Gradient-Forces, Invariant) | Undistilled | 8.4 | **0.08** | | |
| | | | JMP-S Hessian (ours) | 5.0 | 0.09 | | |
| | | | JMP-L Hessian (ours) | **4.0** | 0.1 | **1.4** (3.5x) | **9.6** (16x) |
| | | (S) eSCN (0.94M) (Direct-Forces, $l=2$ Equivariant) | Undistilled | **8.4** | 1.5 | | |
| | | | JMP-S Hessian (ours) | 9.9 | **0.79** | | |
| | | | JMP-L Hessian (ours) | 9.9 | 0.80 | **1.6** (4x) | **13.5** (2.3x) |
| Double Walled Nanotube | 800 | (FM) JMP-S (39.9M) | – | 23.8 | – | 0.5 | 0.7 |
| | | (FM) JMP-L (220M) | – | 11.8 | – | 0.2 | 0.2 |
| | | (S) GemNet-dT (0.67M) (Direct-Forces, Invariant) | Undistilled | 14.3 | 0.49 | | |
| | | | JMP-S Hessian (ours) | 14.3 | **0.23** | | |
| | | | JMP-L Hessian (ours) | **10.6** | 0.25 | **3.2** (16x) | **6.4** (32x) |
| | | (S) GemNet-T (0.57M) (Gradient-Forces, Invariant) | Undistilled | 13.6 | 0.07 | | |
| | | | JMP-S Hessian (ours) | 12.9 | **0.05** | | |
| | | | JMP-L Hessian (ours) | **10.8** | 0.06 | **1.8** (9x) | **3.3** (16.5x) |
| | | (S) eSCN (0.94M) (Direct-Forces, $l=2$ Equivariant) | Undistilled | 19.2 | 0.50 | | |
| | | | JMP-S Hessian (ours) | **16.1** | **0.40** | | |
| | | | JMP-L Hessian (ours) | **16.1** | 0.47 | **2.6** (13x) | **5.4** (27x) |

## 4.3 DISTILLING JMP ON MD22

As a final evaluation of our Hessian distillation approach, we distill JMP (Shoghi et al., 2023) FMs finetuned on the largest molecules in MD22—the buckyball catcher and double-walled nanotube—into various student models. This setting presents a number of unique challenges. The selected MD22 molecules, with 148 and 370 atoms respectively, are significantly larger than those considered thus far. Additionally, the JMP models are considerably larger than the previously considered MACE FMs, with approximately 40M and 220M learnable parameters in the small (JMP-S) and large (JMP-L) models respectively. Unlike their MACE counterparts, the JMP FMs must therefore forgo a gradient-based force parameterization to remain within GPU memory limits. The JMP model is also based on a GemNet backbone, which does not utilize built-in higher-order equivariance like the MACE FMs.

Using JMP-S and JMP-L FMs as teachers, we perform Hessian distillation on the selected MD22 molecules to obtain specialized GemNet-dT, GemNet-T, and eSCN student MLFFs. GemNet-T uses a gradient-based force parameterization, while eSCN uses higher-order ($l=2$) equivariance. We find that the our specialized MLFFs are up to $30\times$ times faster than the original FMs, and considerably outperform the undistilled baselines in Force and Energy MAE on both molecules (Table 3). We highlight that our distillation procedure is effective when distilling from a FM with a direct-force parameterization into a student model with gradient-based forces (GemNet-T) and higher-order equivariance (eSCN). We capitalize on this property by running constant energy (NVE) simulations of the buckyball catcher for 100 ps using the trained GemNet models. We find that distillation produces a GemNet-dT model which conserves energy better than its undistilled counterpart and the original JMP-L FM (Figure 2a). We also find that our GemNet-T student MLFF, employing gradient-based forces and distilled with JMP-L Hessians, is able to conserve energy and simulate stably for the entire duration of the 100 ps simulation, while the JMP-L energy gradually drifts throughout the simulation (Figure 2b). We finally highlight that distilling with Hessians from JMP-L leads to better performance than distilling from JMP-S (see **Distillation Method** column of Table 3), suggesting that continued scaling of FMs has the potential to further improve student model performance.

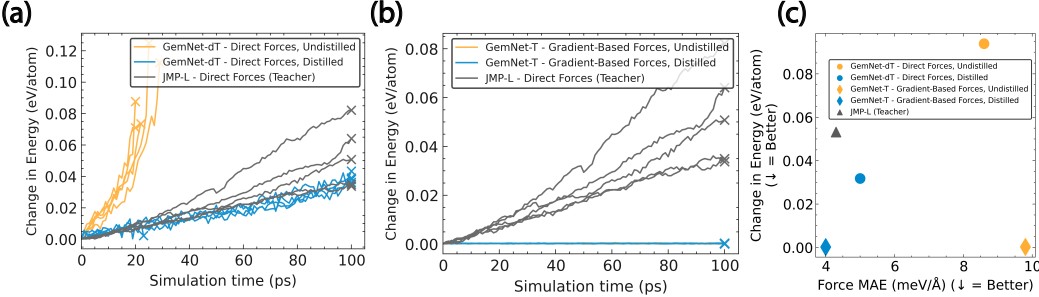

Figure 2: **Energy Conservation in NVE MD Simulations of Buckyball Catcher.** We plot the change in the model predicted energy over the trajectory for 5 independent initial conditions. Some simulations become unstable before 100 ps (denoted by ×). (a) Hessian distillation improves the energy conservation of GemNet-dT models, which outperforms that of JMP-L. (b) Our student GemNet-T models conserve energy due to using conservative forces, while the JMP-L FM energy steadily drifts, broadly suggesting that large-scale models with few built-in constraints can be effectively distilled into smaller, constrained models. (c) Change in energy plotted against test force MAE. Distillation into a GemNet-T student combines the general-purpose representations and accuracy of JMP-L with the physical inductive biases of conservative forces.

## 5 ABLATIONS

We conduct ablation studies on various aspects of our approach, namely: the size of the student MLFF model, and the Hessian subsampling frequency used during training. In §A.9, we additionally examine role of teacher Hessians vs. forces in the KD objective, and find that distilling with teacher forces is significantly inferior to our approach of distilling with Hessians.

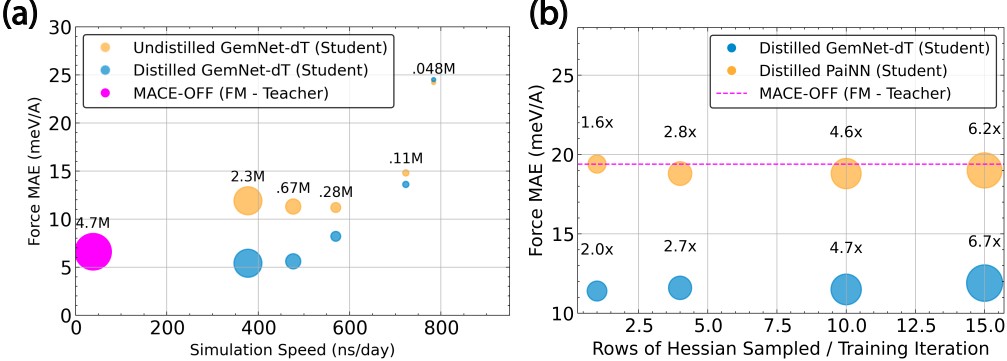

Figure 3: **Parameter count and Hessian subsampling ablations.** (a) Force MAE on the Monomers split of SPICE as a function of the GemNet-dT student MLFF simulation speed. The size of the dots indicates the relative number of trainable parameters in the each model, and the number of parameters is indicated above each dot. Compared to the undistilled model, Hessian distillation improves the speed-accuracy tradeoff. (b) Force MAE on the Solvated Amino Acid split of SPICE as a function of the number of rows of the energy Hessian subsampled at each training iteration. The size of the dots and text indicates the time required per step of training, relative to training without distillation. Reducing down to $s = 1$ does not have a detrimental effect on model accuracy, and results in more efficient training.

**Student MLFF Size.** To understand the effect of student MLFF expressivity, we vary the GemNet-dT student model parameter count by reducing the node and edge embedding dimensions from 128 to 8, which yields a total parameter count varying from 2.3 million to 48,000. For each model, we train with Hessian distillation on the Monomers subset of SPICE. We also train on the same subset without Hessian distillation for comparison. To measure student MLFF speed, we use a larger inference batch size of 2048, which is the point of memory saturation on a NVIDIA A6000 GPU. For the MACE-OFF teacher, we use a batch size of 128, which maximizes its throughput. We find that Hessian distillation significantly improves the trade-off between speed and accuracy at all student MLFF sizes (Fig. 3 a). The differences

in speed between the FM and student MLFFs are more dramatic at larger batch sizes, suggesting that speed benefits from distillation are magnified when parallelizing across more samples. Interestingly, Hessian KD also appears to unlock better scaling properties: without distillation, the force MAE plateaus after scaling to 0.28M parameters, while using Hessian KD yields continual improvements up to the maximum student size of 2.3M parameters. We speculate that the Hessian supervision term may have a regularizing effect on larger models. The improvement becomes more marginal as the student size decreases, indicating that insufficiently expressive students may struggle to minimize the multi-term KD objective.

**Hessian Subsampling Quantity.** We vary the number of rows $s$ sampled from the MACE-OFF FM's reference Hessian during training on the solvated amino acid split of SPICE. We find that increasing $s$ does not reliably lead to a decrease in Force MAE, and in some cases leads to a slight increase. The training cost, measured in GPU-seconds per training step, increases approximately linearly with $s$ (Fig. 3b). With $s = 1$, distillation incurs a $1.6\times$ and $2\times$ increase in training cost relative to undistilled training for PaiNN and GemNet-dT, respectively. We speculate that $s$ may play a similar role as the batch size in conventional training. Small values of $s$ add variance to the KD gradient estimates, facilitating escape from local minima in the loss landscape, while large values may lead to generalization gaps, similar to what has been observed for large batch sizes (Keskar et al., 2016).

## 6 CONCLUSION

**Key Takeaways.** We have presented Hessian distillation, which is to our knowledge the first technique to derive fast, specialized MLFFs by training them to match the energy Hessian of FMs trained on large, diverse datasets. By subsampling rows of the Hessian to supervise the student model at each training iteration, we ensure that Hessian distillation incurs only a nominal cost relative to training the original FM. The specialized student MLFFs derived from our distillation approach are up to $20\times$ faster at predicting energies and forces than the original FMs. Despite having far fewer parameters than the teacher FM, our distilled student models are consistently superior to the FM, as well as models trained with other distillation methods or no distillation. All of the demonstrated improvements readily extend to geometry optimizations and MD simulations, where our distilled models are more stable and conserve energy better over time. Our observation that the student MLFFs often outperform the original FM on the specialized data subset, sometimes even without any distillation, suggests that the field has not yet converged on an effective training recipe for large-scale MLFF FMs, as scaling data and model size should in principle lead to better downstream performance (Kaplan et al., 2020; Hoffmann et al., 2022). We find that our Hessian KD approach still yields improvements over undistilled models in this scenario, suggesting that even when the FM forces are inaccurate, its Hessians still provide effective regularization to make significant improvements in student force accuracy. More generally, the JMP (§4.3) and EScAIP (§A.1) results suggest that Hessian distillation will improve in effectiveness as FMs scale and become more performant.

**Limitations.** The main drawback of our method is that training with Hessian distillation adds training overhead that scales with the number of sampled rows $s$. However, since choosing $s = 1$ has an empirically negligible impact on performance (§5), we can limit the training overhead to around twice that of the undistilled models. Since student models tend to be much smaller than FMs, the training overhead relative to that of the corresponding foundation model is quite small. We also demonstrate that it is possible to accelerate Hessian computation using finite difference approximations in §A.11.

**Future Work and Outlook.** In the future, MLFFs could be specialized in ways beyond chemical subgroups, such as performing high-temperature simulations (Stocker et al., 2022) or modeling phase transitions (Jinnouchi et al., 2019) using Hessian-based KD. Applying sampling techniques when pre-computing the teacher Hessians would also reduce the upfront cost of our approach. More broadly, our work sets a precedent for future MLFF development: as training data and model parameter counts continue to grow, new MLFF FM releases should be accompanied by a set of small, specialized "engines" for common downstream tasks. We further speculate that in the future, practitioners may rarely, if ever, actually perform inference with MLFF FMs directly. Instead, FMs could serve as a reservoir for general-purpose representations, which are subsequently distilled into small models specialized for the task at hand. In particular, the JMP and EScAIP distillation results suggest a recipe in which large FMs are trained with minimal inductive biases to facilitate scalable and general-purpose training, followed by distillation into specialized student models with inductive biases tailored to the downstream task (e.g., conservative forces for MD simulations). This paradigm would enable widespread adoption of MLFFs, and move the field closer to the longstanding dream of force fields with the speed of classical methods and the accuracy of quantum mechanical methods.

## 7 ACKNOWLEDGMENTS

The authors thank Eric Qu for assistance with training and distilling the EScAIP model. We also thank Ryan Liu, Toby Kreiman, Rasmus Lindrup, and the FAIR chemistry team for helpful discussions that benefited this paper. The authors acknowledge the support of this work from the U.S. Department of Energy, Office of Science, Energy Earthshot initiatives as part of the Center for Ionomer-based Water Electrolysis at Lawrence Berkeley National Laboratory under Award Number DE-AC02-05CH11231. This research used the National Energy Research Scientific Computing Center (NERSC), a U.S. Department of Energy Office of Science User Facility located at Lawrence Berkeley National Laboratory, operated under Contract No. DE-AC02-05CH11231 using NERSC award BES-ERCAP0028133.

### 7.1 REPRODUCIBILITY STATEMENT

We have built our implementation of Hessian distillation around the Fairchem Github Repository. Our implementation works with any model or dataset compatible with Fairchem. We plan to release the code after acceptance of the paper. Details on datasets, models, and hyperparmeters used for training and evaluation are provided in the Appendix.

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

## A  APPENDIX

### A.1  RESULTS OF DISTILLING ESCAIP ON SPICE

Table 4: Results of distilling the EScAIP model trained on SPICE into specialized GemNet-T student MLFFs. (FM) indicates foundation model, while (S) indicates student model. **n2n** is the node feature matching baseline from (Kelvinius et al., 2023), while **a2a** is the atom embedding matching baseline we construct in §A.6. Student models all have identical simulation speeds. Speedups relative to FM are given in parentheses. The "Speed" column is calculated with a constant batch size of 64 for all models, while the "Maximum Speed" is calculated with the batch size that maximizes throughput for each model.

| Chemical Subgroup | Dataset Size | Model (Parameter Count) | Distillation Method | Force MAE (meV/Å) (↓) | Energy MAE (meV/atom) (↓) | Speed (ns/day) (↑) | Maximum Speed (ns/day) (↑) |
|---|---|---|---|---|---|---|---|
| Monomers | 14,331 | (FM) EScAIP Large (45M) | – | 3.48 | 0.41 | 22.6 | 22.6 |
| | | (S) GemNet-T (0.67M) | Undistilled | 7.9 | 3.8 | | |
| | | | n2n | 15.0 | 6.5 | | |
| | | | a2a | 8.9 | 1.9 | | |
| | | | Hessian (ours) | **4.8** | **2.9** | **66.0** (2.9x) | **408.9** (18.1x) |
| Solvated Amino Acids | 805 | (FM) EScAIP Large (45M) | – | 11.52 | 0.61 | 19.6 | 19.6 |
| | | (S) GemNet-T (0.67M) | Undistilled | 18.3 | 1.7 | | |
| | | | n2n | 29.2 | 2.7 | | |
| | | | a2a | 17.8 | 1.6 | | |
| | | | Hessian (ours) | **10.0** | **0.9** | **23.4** (1.2x) | **23.4** (1.2x) |
| Systems with Iodine | 11,171 | (FM) EScAIP Large (45M) | – | 11.4 | 0.73 | 22.9 | 22.9 |
| | | (S) GemNet-T (0.67M) | Undistilled | 15.9 | 4.0 | | |
| | | | n2n | 29.1 | 12.3 | | |
| | | | a2a | 18.6 | 3.9 | | |
| | | | Hessian (ours) | **10.9** | **2.2** | **65.4** (2.9x) | **128.0** (5.6x) |

Complementary to the results in Section 4.1 (distilling the MACE-OFF FM on SPICE), we distill the recent EScAIP Qu & Krishnapriyan (2024) model trained on SPICE into specialized GemNet-T student models. Unlike MACE-OFF, EScAIP does not explicitly build in SO(3) equivariance. We use the version of EScAIP that employs a direct force parameterization. As the current state-of-the-art model on SPICE, EScAIP has lower energy and force errors than MACE-OFF on all three chemical subsets we consider. As shown by the results in Tab. 4, we find that this translates to improved results with our Hessian distillation procedure: the GemNet-T student models distilled from EScAIP outperform those distilled from MACE-OFF across all chemical subsets in both force and energy MAEs. While distillation from MACE-OFF to GemNet-T occasionally led to increases in energy MAE relative to the undistilled model, these issues are eliminated when distilling from EScAIP. This underscores the promise of leveraging large-scale, unconstrained foundation model training procedures and subsequently distilling into specialized student MLFFs with the inductive biases required for the downstream task at hand.

### A.2  PHYSICAL INTUITION OF ENERGY HESSIAN

The energy Hessian corresponds to the curvature of the energy surface with respect to atomic displacements. The eigenvalues of the energy Hessian are the squares of the normal mode vibrational frequencies, while the eigenvectors represent the amplitudes of motion along each of the $3N$ mass-weighted Cartesian coordinates associated with each mode (Jensen, 2017). These vibrational frequencies are crucial for understanding the thermodynamic properties of molecules, such as heat capacity, entropy, and free energy (Jensen, 2017). These frequencies can also be observed experimentally (Wilson et al., 1980). Energy Hessians are also directly used in geometry optimization algorithms to relax molecular structures (Fletcher, 2000).

### A.3  FOUNDATION MODELS

For the foundation (teacher) models, we use the publicly available, pretrained weights. For MACE-OFF and MACE-MP, we perform no additional finetuning/modifications. For MACE-OFF, the energy MAE values we obtained using the pretrained checkpoint are slightly higher than what is reported in the original paper (Kovács et al., 2023), while the force MAE values are identical. For the JMP models, we finetune

the publicly available, pretrained weights of both JMP-S and JMP-L on the buckyball catcher and double walled nanotube separately, using the exact configurations and hyperparameters provided in the JMP repository. While we were not able to exactly reproduce the force MAE results reported in the JMP paper (Shoghi et al., 2023) (our obtained losses are slightly higher), we did not perform further tuning or experimentation due to limited computational resources. For EScAIP, we obtained the pretrained SPICE model checkpoint directly from the authors.

We note that MACE-MP-0 was trained on the entirety of MPtrj, and an official held-out test set is not publicly available. We created our own MPtrj train/validation/test splits for the student models. The reported Force MAEs are measured on our created test split, which was seen by the FM model during its training, but *not* by our student models.

Below we provide the links to repositories where the foundation model weights were obtained, as well as the training data:
Mace-OFF repository
Mace-MP repository
JMP repository
Spice Dataset
MPtrj Dataset

We report the foundation model training times in Table A.3 , and selected distillation training times in Table 12, both in GPU hours.

Table 5: Training Times for Foundation Models (trained on A100 GPUs)

| Model | Dataset | GPU hours |
|---|---|---|
| MACE-OFF Large | SPICE | 336 (Kovács et al., 2023) |
| MACE-MP Large | MPtrj | 1920 (Batatia et al., 2023) |
| JMP-Small (Pretraining) | QM9 + ANI-1x + OC20 + Trans1x | 5700 (Shoghi et al., 2023) |
| JMP-Small (Finetuning) | Buckyball Catcher | 9 (trained ourselves) |
| JMP-Small (Finetuning) | Double Walled Nanotube | 20 (trained ourselves) |
| JMP-Large (Pretraining) | QM9 + ANI-1x + OC20 + Trans1x | 34400 (Shoghi et al., 2023) |
| JMP-Large (Finetuning) | Buckyball Catcher | 15 (trained ourselves) |
| JMP-Large (Finetuning) | Double Walled Nanotube | 41 (trained ourselves) |

## A.4 STUDENT MLFF ARCHITECTURE DETAILS

We provide details on the architectures of the GemNet-dT, GemNet-T, PaiNN, and eSCN MLFFs used as student models in this work. A slashed value indicates that different values were used across datasets. In this case, the values correspond to the ordering: SPICE, MD22, MPtrj. Since PaiNN was only used for SPICE and MPtrj, in that case the ordering is SPICE, MPtrj.

Table 6: Hyperparameters for PaiNN student models.

| Parameter | Value |
|---|---|
| Hidden Channels | 128 |
| Layers | 4 |
| Radial Basis Functions | 128 |
| Cutoff | 12.0 / 6.0 |
| Maximum Neighbors | 50 |

Within a dataset, hyperparameters for GemNet-dT and GemNet-T models are identical. The only difference is that GemNet-T computes forces as the negative gradient of the energy prediction, while GemNet-dT employs a direct force parameterization.

Table 7: Hyperparameters for GemNet-dT and GemNet-T student models.

| Parameter | Value |
|---|---|
| Number of Spherical | 7 |
| Radial Basis Functions | 6 |
| Blocks | 4 |
| Atom Embedding Size | 64 |
| Edge Embedding Size | 64 |
| Triplet Embedding Size | 32 |
| RBF Embedding Size | 16 |
| CBF Embedding Size | 16 |
| Bilinear Triplet Embedding Size | 64 |
| Number Before Skip | 1 |
| Number After Skip | 1 |
| Number of Concatenations | 1 |
| Number of Atoms | 2 |
| Cutoff | 5.0 / 5.0 / 6.0 |
| Maximum Neighbors | 50 |
| RBF Function | Gaussian |
| Envelope Function | Polynomial (Exponent: 5) |
| CBF Function | Spherical Harmonics |
| Output Initialization | HeOrthogonal |
| Activation Function | SiLU |

## A.5 STUDENT MLFF TRAINING DETAILS

We use the FAIR-chem repository `https://github.com/FAIR-Chem/fairchem`, implemented with PyTorch, for model training and evaluation. We include the training details below.

Table 8: Optimization hyperparameters for student models.

| Parameter | GemNet-dT/GemNet-T/eSCN | PaiNN |
|---|---|---|
| Initial Learning Rate | 0.001 | 0.001 |
| Optimizer | AdamW | AdamW |
| Weight Decay | 0.000002 | 0.000002 |
| Amsgrad | True | True |
| Adam epsilon | 1.e-7 | 1.e-7 |
| Scheduler | ReduceLROnPlateau | ReduceLROnPlateau |
| Patience | 5 | 10 |
| Factor | 0.8 | 0.8 |
| Minimum Learning Rate | 0.000001 | 0.000005 |
| EMA Decay | 0.999 | 0.999 |
| Clip Gradient Norm | 10 | 10 |

We chose batch size and number of Hessian rows primarily based on dataset and system sizes and how these would affect training times.

We report training times for our Hessian distillation approach as a percentage of the original foundation model training time. The caveat is that MACE foundation models were trained on faster, NVIDIA RTX A100 GPUs, while the disilled runs were trained on slower NVIDIA RTX A6000 GPUs. Therefore, these percentages are likely overestimates.

## A.6 BASELINES

We provide additional details on the **n2n** and **a2a** baselines against which we compare our Hessian distillation approach.

Table 9: Loss Weights by Chemical Subset. The same energy and force weights are used for undistilled and distilled training, as well as all baselines.

| Training Set | $\lambda_U$ | $\lambda_F$ | $\lambda_{KD}$ | $\lambda_{\nabla U}$ |
|---|---|---|---|---|
| Monomers | 5 | 100 | 400 | 5 |
| Solvated Amino Acids | 5 | 100 | 400 | 5 |
| Structures with Iodine | 5 | 100 | 400 | 5 |
| $Pm\bar{3}m$ Spacegroup | 0 | 100 | 200 | 0 |
| Structures with Yttrium | 0 | 100 | 200 | 0 |
| Bandgap $\geq$ 5meV | 0 | 100 | 200 | 0 |
| Buckyball Catcher | 5 | 100 | 400 | 5 |
| Double Walled Nanotube | 5 | 100 | 400 | 5 |

Table 10: Training Batch Size for Student Models by Chemical Subset. The same batch sizes are used for undistilled and distilled training.

| Training Set | GemNet-dT/GemNet-T | PaiNN | eSCN |
|---|---|---|---|
| Monomers | 4 | 8 | – |
| Solvated Amino Acids | 4 | 8 | – |
| Structures with Iodine | 4 | 8 | – |
| $Pm\bar{3}m$ Spacegroup | 16 | 16 | – |
| Structures with Yttrium | 16 | 16 | – |
| Bandgap $\geq$ 5meV | 32 | 32 | – |
| Buckyball Catcher | 4 | – | 4 |
| Double Walled Nanotube | 4 | – | 4 |

Table 11: Number of rows sampled from Hessian

| Training Set | GemNet-dT/GemNet-T | PaiNN | eSCN |
|---|---|---|---|
| Monomers | 4 | 4 | – |
| Solvated Amino Acids | 1 | 1 | – |
| Structures with Iodine | 4 | 4 | – |
| $Pm\bar{3}m$ Spacegroup | 4 | 4 | – |
| Structures with Yttrium | 1 | 4 | – |
| Bandgap $\geq$ 5meV | 1 | 4 | – |
| Buckyball Catcher | 1 | – | 1 |
| Double Walled Nanotube | 1 | – | 1 |

Table 12: Training Times in GPU-hours for Selected Distilled Runs. Percentages indicate the fraction of time the training run took compared to the training time of the relevant foundation model.

| Training Subset | GemNet-dT | PaiNN |
|---|---|---|
| Monomers | 72.5 (21.5%) | 30.5 (9.1%) |
| Solvated Amino Acids | 15.2 (4.5%) | 9.3 (2.8%) |
| Structures with Iodine | 68.3 (20.3%) | 29.8 (8.8%) |
| $Pm\bar{3}m$ Spacegroup | 14.8 (0.8%) | 7.5 (0.4%) |
| Structures with Yttrium | 57.1 (3.0%) | 65.0 (3.4%) |
| Bandgap $\geq$ 5meV | 82.2 (4.3 %) | 48.7 (2.5%) |
| Buckyball Catcher | 87 (0.25 %) | – |
| Double Walled Nanotube | 144 (0.42 %) | – |

**Node feature supervision (n2n).** The n2n approach, introduced in (Kelvinius et al., 2023), seeks to align the node features of the student MLFF with that of the teacher. Specifically, given node features $h_T^{(l)} \in \mathbb{R}^{d_t}$ and $h_S^{(l)} \in \mathbb{R}^{d_s}$ from the $l^{th}$ message-passing layer of the teacher and student respectively, the

distillation loss is formulated as

$$L_{\text{KD}} = \mathbb{E}_x ||h_T^{(l)}(x) - Ph_S^{(l)}(x)||_2^2,$$

where $P \in \mathbb{R}^{d_t \times d_s}$ is a learnable linear projection from the student to teacher representation space. The projection weights are discarded at inference time. (Kelvinius et al., 2023) found that that using the final layer node representation ($l = L$) yielded the best results, so we chose the same. As in the Hessian distillation setting, we pre-compute and save the teacher's final node features over the dataset prior to training. We found via a sweep over $\lambda_{KD} = \{10, 100, 1000, 10000, 100000\}$ that $\lambda_{KD} = 10000$ yields the best results with GemNet-dT on the Solvated Amino Acid split of SPICE (results shown below). Due to a limited computed budget, we do not perform sweeps over each individual split, and use this same value for all subsequent splits and models.

Table 13: Validation energy MAE (meV) and force MAE (meV/A) of the **n2n** baseline on the Solvated Amino Acid split of SPICE, using different values of $\lambda_{KD}$. Energy MAE is total MAE, so there is not a one-to-one correspondence between the per-atom MAE results reported in the main text.

| $\lambda_{KD}$ | Force MAE (meV/A) | Energy MAE (meV) |
|---|---|---|
| 0 (Undistilled) | 22.4 | 162 |
| 10 | 22.1 | 163 |
| 100 | 22.1 | 169 |
| 1000 | 21.7 | 142 |
| 10000 | **21.1** | **117** |
| 100000 | 28.1 | 144 |

**Atom embedding initialization (a2a).** In GNN-based MLFFs, the initial node features $h^{(0)}$ are parameterized as a learnable dictionary of embeddings for each atomic element. In the a2a approach, we precompute the teacher's atom embeddings $h_T^{(0)}$, and parameterize the student atom embeddings as $h_S^{(0)} = Ph_T^{(0)}$, where $P \in \mathbb{R}^{d_s \times d_t}$ is a learnable linear projection from the teacher to student representation space. We train the MLFF using the original energy/force matching objective, with no distillation (i.e $\lambda_{KD} = 0$). There are no hyperparameters to tune for the **a2a** baseline.

## A.7 EVALUATION DETAILS

**Reporting of Maximum Simulation Speed.** We calculate maximum simulation speeds by performing energy/force inference with the batch size that maximizes throughout for each model. We report these batch sizes below.

Table 14: Throughput-maximizing batch sizes

| Training Set | GemNet-dT | PaiNN | MACE | GemNet-T | eSCN | JMP-S | JMP-L |
|---|---|---|---|---|---|---|---|
| Monomers | 1024 | 1024 | 128 | – | – | – | – |
| Solvated Amino Acids | 64 | 64 | 32 | – | – | – | – |
| Structures with Iodine | 512 | 512 | 64 | – | – | – | – |
| $Pm\bar{3}m$ Spacegroup | 512 | 512 | 512 | – | – | – | – |
| Structures with Yttrium | 128 | 512 | 128 | – | – | – | – |
| Bandgap $\geq$ 5meV | 64 | 64 | 64 | – | – | – | – |
| Buckyball Catcher | 64 | – | – | 24 | 32 | 4 | 4 |
| Double Walled Nanotube | 8 | – | – | 8 | 12 | 4 | 4 |

We convert inference speed from samples/second to nanoseconds/day by adopting a MD simulation timestep of 1 femtosecond, and assuming that energy/force inference dominates simulation time.

## A.8 SENSITIVITY TO KNOWLEDGE DISTILLATION WEIGHT

We assess the sensitivity of our Hessian distillation approach to the knowledge distillation loss weight $\lambda_{KD}$ used during training. We vary the weight across $\lambda_{KD} = \{10, 100, 400, 1000\}$ on the Solvated Amino

Acids split of SPICE with a GemNet-dT model. We find that a value of $\lambda_{KD} = 400$ achieves the best balance of force and energy performance (full results below).

Table 15: Validation energy MAE (meV) and force MAE (meV/A) achieved by Hessian distillation on the Solvated Amino Acid split of SPICE, using different values of $\lambda_{KD}$. Energy MAE is total MAE, so there is not a one-to-one correspondence between the per-atom MAE results reported in the main text.

| $\lambda_{KD}$ | Force MAE (meV/A) | Energy MAE (meV) |
|---|---|---|
| 0 (Undistilled) | 22.4 | 162 |
| 10 | 20.2 | **145** |
| 100 | 14.0 | 168 |
| 400 | **12.3** | 155 |
| 1000 | 13.7 | 181 |

We performed a similar sweep for MPTrj and found that $\lambda_{KD} = 100$ was optimal in that setting. Increasing the KD weight generally helps up to a certain point, after which performance saturates and eventually degrades. This is an important hyperparameter that we recommend be tuned for each dataset independently.

## A.9 EFFECT OF TEACHER HESSIANS VERSUS FORCES.

To isolate the benefit of the FM Hessians, we formulate an alternative knowledge distillation objective where the student is trained to match the FM force predictions instead of its energy Hessians. We simply replace the ground truth forces in Eq. 1 with the foundation model (teacher) forces with a weight of $\lambda_F = 100$ (the same weight previously used for the ground truth forces). Concretely, the new loss function becomes,

$$\mathcal{L} = \lambda_U |U_{\text{ref}}(\mathbf{z},\mathbf{r}) - U_\theta(\mathbf{z},\mathbf{r})|^2 + \lambda_F \sum_{i=1}^{n} \|\mathbf{f}_{\text{FM}}^{(i)}(\mathbf{z},\mathbf{r}) - \mathbf{f}_\theta^{(i)}(\mathbf{z},\mathbf{r})\|_2^2 \tag{4}$$

where $\mathbf{f}_{\text{FM}}$ denotes the teacher forces and as per usual, $U_{\text{ref}}$ denotes the ground truth energies.

Results on selected splits of the SPICE dataset are presented in Tab. 16. Distilling with the teacher forces is consistently inferior to distilling with the teacher Hessians, indicating that the richer information contained in the latter helps the model better match the true forces. In fact, force distillation leads to worse performing models than training without distillation, similar to the observation that the **n2n** and **a2a** baselines were inferior to undistilled training on SPICE (§4.1).

Table 16: Ablation study looking at distilling with MACE-OFF forces on selected splits of the SPICE dataset. This approach is consistently inferior to our approach of distilling with MACE-OFF Hessians.

| Chemical Subgroup | Student Model | Distillation Method | Force MAE (meV/A)(↓) |
|---|---|---|---|
| Monomers | GemNet-dT | Undistilled | 11.3 |
| | | Forces | 14.2 |
| | | Hessian (ours) | **6.3** |
| | PaiNN | Undistilled | 25.0 |
| | | Forces | 25.0 |
| | | Hessian (ours) | **8.77** |
| Systems with Iodine | GemNet-dT | Undistilled | 23.4 |
| | | Forces | 26.1 |
| | | Hessian (ours) | **14.7** |
| | PaiNN | Undistilled | 51.2 |
| | | Forces | 51.7 |
| | | Hessian (ours) | **23.7** |

## A.10 COMPARISON OF HESSIAN DISTILLATION TO CONSERVATIVE FORCE TRAINING

We compare the benefits obtained from Hessian distillation of direct force student models (GemNet-dT) with training gradient-based MLFFs (GemNet-T) without distillation. Results on SPICE splits are shown in Tab. A.10.

Table 17: Force MAE (meV/A) of conservative force training (GemNet-T) on SPICE test splits, as compared with undistilled and distilled training with GemNet-dT.

| Chemical Subset | GemNet-dT (Undistilled) | GemNet-T (Undistilled) | GemNet-dT (Distilled) |
|---|---|---|---|
| Monomers | 11.3 | 8.8 | **6.3** |
| Solvated Amino Acids | 22.4 | 31.0 | **11.6** |
| Systems with Iodine | 22.6 | 16.0 | **14.7** |

Although training with gradient-based forces yields benefits over training with direct forces in the undistilled setting on 2 out of the 3 chemical subsets, we find that the improvements are less than those achieved by Hessian distillation. We hypothesize that while the inductive bias of conservative forces is beneficial, the extra supervision provided by Hessian distillation is a stronger learning signal to learn on chemical subsets with potentially limited data. We note that the improvements from adding conservative forces incurs roughly a $3 \times$ increase in inference time due to the extra backpropagation step to compute forces, while the distilled GemNet-dT model has the identical architecture as the undistilled model and incurs no such cost. However, for applications such as constant energy (NVE) MD simulations, gradient-based forces may be required to produce physical and stable simulations. We thus combine both of these elements in §4.1 and §4.3 and distill with a gradient-based GemNet-T student model to yield greater force MAE improvements while maintaining test-time energy conservation

## A.11 COMPUTING HESSIANS WITH FINITE DIFFERENCES

In situations where computing the Hessian via autodifferentiation is unfeasible, such as for extremely large foundation models (FMs), models employing attention kernels that are not twice-differentiable (Lefaudeux et al., 2022), or conservative student models where training would require 3rd order gradients, a finite difference approach can be used instead.

In this scheme, molecular structures are perturbed in Euclidean space and energy/force derivatives are obtained via a discretized stencil (in this case, a right difference scheme). While this work focuses on cases where autodifferentiation is feasible, we also demonstrate that the finite difference approach is a viable alternative. We provide results on the Solvated Amino Acids split in Tab. 18 as a proof of concept, showing that the method accelerates training for conservative force student models. The difference in final Force MAE achieved by the two methods is negligible.

Table 18: Computing Hessian with Autodifferentiation vs Finite Differences with GemNet-T (conservative forces) on Solvated Amino Acids.

| Computation Method | Training Speed (Epoch/Min) |
|---|---|
| Autodifferentiation | 1.27 |
| Finite Differences | **1.67** |

## A.12 EFFECTS OF ADDING A LOSS SCHEDULER TO HESSIAN DISTILLATION

While training via Hessian distillation, we reduce the distillation loss coefficient $\lambda_{KD}$ by 1/2 when the student model exceeds the teacher model's accuracy. This approach dynamically adjusts the training process to focus more on matching the ground truth energies and forces rather than the foundation model's Hessians. Below, we compare Hessian distillation with and without the scheduler on a dataset, demonstrating that the scheduler provides small but consistent improvements in Force MAE. We apply the same scheduling to the **n2n** and **a2a** baselines, but in practice the baselines never exceed the teacher accuracy, so the scheduling criterion is not met.

Table 19: Comparison of Hessian Distillation with and without a Loss Scheduler, training with GemNet-dT on Solvated Amino Acids.

| Scheduler | Force MAE |
|---|---|
| Without Scheduler | 12.2 |
| With Scheduler | **11.6** |

### A.13 Isolating the Effect of Hessian Distillation on Energy Accuracy

It is of interest to see how Hessian distillation alone, without our energy gradient supervision term described in §3.4, affects energy accuracy. Below in Tab. 20, we compare a training run using only Hessian distillation with one that combines Hessian and energy gradient supervision. While Hessian distillation alone improves Energy MAE, the addition of energy gradient supervision leads to even greater improvements.

Table 20: Ablation investigating the effects of the Hessian and Energy gradient supervision terms when training PaiNN on Solvated Amino Acids.

| Distillation Method | Energy MAE |
|---|---|
| Undistilled | 3.3 |
| Hessian Distillation only | 2.9 |
| Hessian Distillation + Energy Gradient Supervision | **0.41** |

### A.14 NVT MD Simulations with Student Models

To further evaluate the distilled, specialized MLFFs from §4.1, we run 100 picosecond, constant-temperature (NVT) MD simulations with systems from the Solvated Amino Acid split, with molecules containing 79-96 atoms each. We choose 5 random structures from the held-out test set as initial conditions. We perform Langevin dynamics at a temperature of 300K, a timestep of 1.0 $fs$, and a friction coefficient of $0.01$ $fs^{-1}$, for 100,000 steps, corresponding to 100 ps. Consistent with (Fu et al., 2022), we use a maximum bond length deviation metric, which measures unphysical bond stretching or collapse, to measure stability. According to this criterion, a simulation becomes unstable at time $T$ if,

$$\max_{(i,j)\in\mathcal{B}}|(\|r_i(T)-r_j(T)\|-b_{i,j})|>\Delta,$$

where $\mathcal{B}$ is the set of all bonds, $i,j$ are the two endpoint atoms of the bond, and $b_{i,j}$ is the equilibrium bond length computed from the training dataset. Following (Fu et al., 2022), we set $\Delta = 0.5A$. Since we are simulating non-reactive systems at ambient conditions, bond deviations exceeding this amount are indicative of simulation failure.

Results are shown in Fig. 4. We find that the improvements in Force MAE between the distilled and undistilled MLFFs shown in Tab. 1 translate to considerably improved stability over time. We reiterate that both the GemNet-dT and PaiNN student models lack a conservative force parameterization, which is generally considered crucial for stable MD simulations. Our results thus may indicate that distillation from a FM teacher employing conservative forces may be sufficient to achieve stable simulation without this inductive bias. We leave a more complete analysis of the performance of distilled MLFFs in MD simulations, including the capturing of observables over long timescales (Fu et al., 2022; Raja et al., 2024), for future work.

### A.15 Geometry Optimization with Student Models

As an additional evaluation of the usefulness of our student MLFFs, we perform geometry optimization with our GemNet-dT student models using the FIRE (Bitzek et al., 2006) optimizer in the Atomic Simulation Environment (ASE). We select 100 structures from the Monomers split of the SPICE dataset, and run optimization until all the per-atom force norms are below 0.05 eV/A. Finally, we compute the energy and per-atom forces using Density Functional Theory (DFT) at the $\omega$B97M-D3BJ/def2-TZVPPD level of theory (the same level used to generate the dataset in (Eastman et al., 2023)). We run DFT calculations with the default settings in the psi4 Python package (Turney et al., 2012).

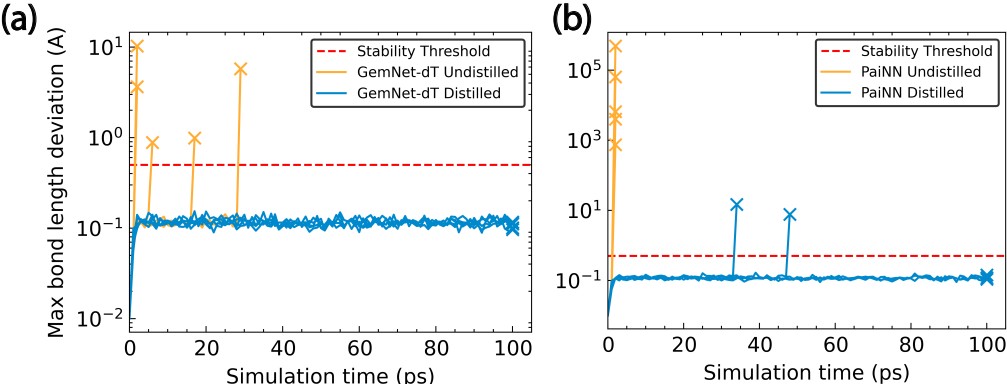

Figure 4: **Stability of Constant Temperature MD Simulations.** Results of constant temperature (NVT) MD simulations using the distilled GemNet-dT and PaiNN student MLFFs. We plot the maximum bond length deviation during NVT simulations of 5 selected systems from the SPICE Solvated Amino Acid split. $\times$ denotes the point at which the simulation becomes unstable. Our distilled models are considerably more stable than their undistilled counterparts, both for (a) GemNet-dT and (b) PaiNN.

We find that our distilled GemNet-dT model generally converges to structures with lower energy and mean per-atom force norms than its undistilled counterpart. While the FIRE optimizer does not explicitly use energy Hessians, many quasi-Newton algorithms like BFGS (Bitzek et al., 2006) do. It would be interesting to explore whether our Hessian distillation approach leads to even greater gains for these optimizers.

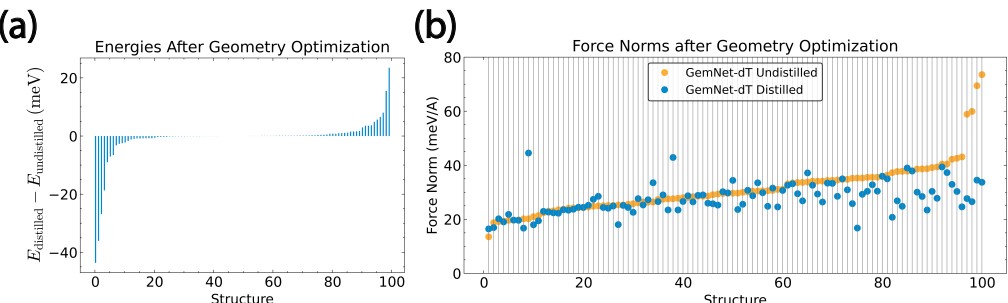

Figure 5: **Geometry optimization with GemNet-dT student MLFFs.** (a) Difference in energy of the final, relaxed structure obtained via the distilled and undistiled models. On average, the distilled model converges to lower energy structures. (b) Mean per-atom force norm of the final, relaxed structure obtained via the distilled and undistiled models. On average, the distilled model converges to lower structures with lower force norms.

