# OpenReview forum: "Towards Fast, Specialized Machine Learning Force Fields: Distilling Foundation Models via Energy Hessians"
_ICLR.cc/2025/Conference — ICLR 2025 Poster_

### Official Review · Reviewer_8TDr · 2024-10-26

**Soundness:** 3
**Presentation:** 3
**Contribution:** 2
**Rating:** 8
**Confidence:** 4

**Summary:**

This paper presents a novel technique for knowledge distillation from foundation model force fields to smaller, faster, and more specialized force fields. The core idea is to align the Hessians of the energies with respect to atomic positions between the teacher (foundation model) and student models, facilitating efficient knowledge transfer. Experiments demonstrate that this approach improves stability and accuracy in the derived force fields compared to simpler distillation methods.

**Strengths:**

1. Clarity and Readability: The paper is very well-written and accessible, making the proposed method easy to follow.
2. Practicality: The approach is straightforward to implement and is cost-effective relative to similar knowledge distillation methods.
3. Experimental Validation: MD simulations performed validate the method, underscoring the practical benefits of the proposed technique.
4. Implementation Insights: The paper also offers practical implementation guidance for practitioners, which is particularly useful for real-world applications.

**Weaknesses:**

The baseline methods used for comparison seem to perform notably poorly, raising questions about fairness. This might be genuine, but the absence of hyperparameter tuning for the baselines undermines this. The authors specifically tuned their method by adjusting the Hessian distillation loss term, "we reduce the weight of the Hessian distillation loss term, λKD, by a factor of 4 during training once the student’s validation loss on the original objective, LEF (φ), becomes lower than that of the frozen FM, LEF (ψ)" (l.207-209). Further clarification on the role of this schedule, as well as the sensitivity to λKD, would be helpful. Would similar tuning or scheduling benefit the alternative approaches as well?

**Questions:**

* Why is there a tank in Figure 1?
* While the Hessian has a nice physical interpretation, as the authors point out, do higher derivates improve transfer further? One way to improve runtime in such cases would be to use Forward on Backward differentiation.

---

> ### Author Response · Authors · 2024-11-20
> **Response**
>
> **Question: Poor performance/lack of tuning of baselines.**
>
> We have performed a sweep of the knowledge distillation weight for the n2n baseline (details in Appendix Section A.4) and used the setting that maximizes performance in the reported results. This has led to improvements in the performance of n2n, which now consistently outperforms undistilled training but is still clearly inferior to our approach. Notably, n2n does improve energies more dramatically than forces, which is consistent with the results reported in Kelvinius, et. al [1].
> The a2a baseline does not have any readily obvious hyperparameters to tune (it has the same loss function as undistilled training).
>
> [1] Ekström Kelvinius, Filip, et al. "Accelerating molecular graph neural networks via knowledge distillation." Advances in Neural Information Processing Systems 36 (2024).
>
> **Question: Role of loss coefficient scheduling and applicability to baselines.**
>
> The intuition between this scheduler is that once the student model becomes better than the teacher (measured by, for example, Force MAE), matching the teacher Hessians likely becomes a less useful training objective. At this point, it makes sense to focus more heavily on matching the ground truth energies and forces, hence the adjustment of the KD loss coefficient. We have added an experiment to Appendix A.10 ablating the loss coefficient scheduler, which gives a slight improvement in Force MAE over a training run not using the scheduler. This trick is in principle also applicable to the baselines, but the validation loss of the n2n and a2a baselines does not ever become lower than that of the frozen FM, so the trick does not apply in practice.
>
> **Question: Sensitivity to knowledge distillation weight.**
>
> We have included an experiment examining the effect of the knowledge distillation weight in Appendix Section A.6. Increasing the KD weight generally improves the Force MAE up to a certain point, after which performance saturates and eventually degrades. This is an important hyperparameter that we recommend be tuned for each dataset independently.
>
> **Question: Tank in Figure 1**
>
> This was meant to highlight that the foundation model is “slow but powerful”, in contrast with the lightning bolt representing fast, specialized student models. We have removed this symbol as it has caused confusion.
>
> **Question: Inclusion of higher-order derivatives.**
>
> While higher-order derivatives could in principle improve performance, we do not include them in this work as they are very computationally expensive to compute, particularly for large foundation models with ~10^7 parameters. Additionally, the total number of components in higher order derivatives scales as n^k, where n is the dimension of the input and k is the order of the derivative. The number of independent components also scales unfavorably as  {(n - 1 + k) \choose k}. Computing all of these components for the teacher model would be extremely expensive, and sampling would likely be much less efficient during training. We also note that truncating at second derivatives has a precedent in many commonly used physics models, including harmonic oscillators and potentials near critical points.
>
> **Question: Forward-on-Backward differentiation.**
>
> As per your suggestion, we did try to implement our Hessian distillation scheme with Forward-on-Backward differentiation to see if it would speed up training for our conservative student models. Unfortunately, this is complicated by the fact that many MLFF architectures use in-place operations, which are incompatible with the relevant torch functional transformations, like jvp, etc. However, another way to speed up Hessian computation is by using finite differences in place of automatic differentiation. When we compute the Hessian using autograd, backpropagating through a conservative model to update the model parameters requires a third order derivative, which for some models might be extremely expensive. Computing the Hessian via finite differences removes an order. We demonstrate in Appendix Section A.9 that we can use a simple right-difference scheme and achieve a 40% speedup over autograd with no degradation in performance on the Solvated Amino Acids spit of SPICE.

---

> > ### Comment · Reviewer_8TDr · 2024-11-24
> >
> > I thank the authors for their thorough response and increase my score accordingly.

---

> > > ### Author Response · Authors · 2024-11-24
> > >
> > > Thank you for your feedback and response!

---

### Official Review · Reviewer_yqHS · 2024-10-28

**Soundness:** 3
**Presentation:** 3
**Contribution:** 3
**Rating:** 6
**Confidence:** 5

**Summary:**

This paper introduces a method to distillate MLFF foundation models to smaller, faster MLFFs with energy Hessians, achieving significant speed improvements while maintaining performance.

**Strengths:**

* This paper proposes a new method to distill MLFF foundation models into smaller, faster MLFFs with Hessians, which is highly beneficial for simulating realistic systems.
* The paper is written in a clear and concise manner, facilitating effortless understanding.

**Weaknesses:**

Related concerns are discussed in the questions section.

**Questions:**

* The tables in the paper show only force results, without energy, so I'm curious about the energy results after distillation.
* A major concern is that the primary use of the distilled MLFF model is for molecular dynamics simulations, where conservation properties are crucial for scientists in physics, chemistry, biology, and materials science. I understand the authors avoided second-order derivatives to calculate the Hessian by directly predicting forces, using JVP calculations. However, a pretrained model might predict forces directly to save computation due to its large size, but the student model should compute forces using autograd, similar in the SFT in JMP[1], which makes more sense. Although Fig. 2 shows stable molecular dynamics in the NVT ensemble, following [2], energy will not be conserved in the NVE ensemble.
* The paper claims that the student model outperforms the teacher model, which is confusing. I suspect this is because the energy and force labels used in training come from the dataset itself. While the inclusion of Hessian loss is shown to be better than using only energy and force loss, this highlights the importance of derivatives. Since the Hessian matrix introduces force derivatives, could training a traditional MLFF from scratch, with forces computed via autograd, achieve similar or better results? Additionally, the statement in Fig. 3b about "speculating that s may play a similar role as the batch size" is akin to conclusions from traditional MLFF training, suggesting that direct autograd training without Hessian distillation might yield similar outcomes. The authors could compare such models to illustrate the Hessian's impact.
* Regarding the appendix experiment using force for distillation, what is the specific loss function? If I'm correct in understanding that the Hessian term in Eq. 3 is replaced by the force term, could the poor results from force distillation be due to the inherent force label loss in the data, where the teacher model's force predictions contradict the data labels? This implies that the force distillation setup might be flawed.

[1] Shoghi N, Kolluru A, Kitchin J R, et al. From Molecules to Materials: Pre-training Large Generalizable Models for Atomic Property Prediction[C]//The Twelfth International Conference on Learning Representations.

[2] Fu X, Wu Z, Wang W, et al. Forces are not Enough: Benchmark and Critical Evaluation for Machine Learning Force Fields with Molecular Simulations[J]. Transactions on Machine Learning Research.

---

> ### Author Response · Authors · 2024-11-20
> **Response**
>
> **Question: Energy results after distillation.**
>
> We have updated Table 1 to include energy results on the SPICE dataset. We achieve significant improvements in energy by utilizing Hessian distillation, as well as an additional loss term based on the gradient of the energy head of non-conservative student models, described in Section 3.4. Both the Hessian Distillation and the new loss term contribute to improvements in the energy MAE, as shown in the ablation in Appendix Section A.11.
>
> **Question: Energy-conserving student models.**
>
> Thank you for raising this very important point. Please refer to Section 4.3 (“Distilling JMP on MD22”) of the updated paper. We have included a series of new experiments where we use JMP, a large foundation model which uses non-conservative forces, as the teacher model, and GemNet-T, which uses conservative force, as the student model. We find that the Hessian distillation approach works well in this setting, leading to improvements in Force MAE over undistilled baselines (Table 3). Crucially, we also find that the JMP-L foundation model does not conserve energy well in NVE MD simulations, while our distilled GemNet-T models do so by construction and produce stable simulations (Figure 2). This highlights the potential usefulness of distilling into student models with inductive biases suited to the downstream task at hand, even if the foundation model lacks these inductive biases. Also note the related point we have added in the Conclusion: “...the energy conservation results in Section 4.3 suggest a recipe in which large FMs are trained with minimal inductive biases to facilitate scalable and general-purpose training, followed by distillation into specialized student models with inductive biases tailored to the downstream task (e.g. conservative forces for constant energy MD simulations).”
> We found that distilling into conservative student models was reasonably fast despite requiring third-order gradients - two for the Hessian calculation, and one for optimizing the loss - see Table 11 in the Appendix for exact training times. If training time does become an issue in the future (e.g. with larger student models), we demonstrate in Appendix Section A.9 that computing Hessians with finite differences is a viable alternative to auto-differentiation that yields nearly a 40% speedup in training without sacrificing accuracy.
>
> **Question: Comparison to traditional MLFFs with forces computed via autograd.**
>
> We have run an ablation study on this by training a GemNet-T student model, which computes forces using autograd, without Hessian distillation on the chemical subsets of SPICE. The results have been added to Table 16 in Appendix section A.8: “Comparison of Hessian Distillation to Conservative Force Training.” We find that while training an undistilled, gradient-force GemNet-T student model generally yields improvements over an undistilled, direct-force GemNet-dT model, the improvements are not as large as those achieved by Hessian distillation. We hypothesize that while the inductive bias of conservative forces is beneficial, the extra supervision provided by Hessian distillation is a stronger learning signal to learn on chemical subsets with potentially limited data. We also note that we could always perform distillation with student MLFFs which compute forces via autograd, combining the best of both worlds. We have shown that this is possible on MD22 with JMP as a teacher model (see Section 4.3 and previous response).
>
> **Question: Appendix experiment with teacher force distillation.**
>
> We apologize for the confusion. The correct way to describe this experiment is that we replace the ground truth forces in Eqn. 1 (the standard energy/force matching loss) with the forces computed by the teacher model. Therefore, we do not have an issue with the teacher labels contradicting the data labels. We have corrected the explanation and included the loss function (Eqn 4) in the Appendix (Section A.7) .
>
> [1] Shoghi N, Kolluru A, Kitchin J R, et al. From Molecules to Materials: Pre-training Large Generalizable Models for Atomic Property Prediction[C]//The Twelfth International Conference on Learning Representations.
>
> [2] Fu X, Wu Z, Wang W, et al. Forces are not Enough: Benchmark and Critical Evaluation for Machine Learning Force Fields with Molecular Simulations[J]. Transactions on Machine Learning Research.

---

> > ### Comment · Reviewer_yqHS · 2024-11-25
> >
> > Thank you for the response. The additional experiments you conducted, particularly "Distilling from a non-conservative teacher model to a conservative or higher-order equivariant student model," have addressed my concerns. I am willing to increase my score.

---

### Official Review · Reviewer_8NSg · 2024-11-02

**Soundness:** 2
**Presentation:** 3
**Contribution:** 2
**Rating:** 6
**Confidence:** 3

**Summary:**

This paper introduces a method for transferring general-purpose representations from large ML force field (MLFF) foundation models to smaller, faster MLFFs specialized for specific regions of chemical space, with the aim of improving inference speed. The approach is formulated as a knowledge distillation (KD) process, where the smaller “student” MLFF learns to match the Hessians of energy predictions made by the “teacher” foundation model. By selectively subsampling rows of the Hessian corresponding to individual atomic coordinates, the “student” MLFF achieves a training process that is computationally efficient relative to foundation models and demonstrates improved force prediction on downstream datasets.

**Strengths:**

* The approach is well-motivated, leveraging knowledge from large MLFF foundation models and adapting it to specific chemical space regions using knowledge distillation. The method also achieves promising results across organic and material molecular systems.
* The paper is well-organized and easy to follow.

**Weaknesses:**

* Although the method shows promising results, the rationale for using Hessian information as a distillation signal is unclear, which may impact the perceived technical contribution. Additional theoretical or intuitive insights on this choice would clarify the method’s grounding.
* Foundation models are often trained with energy and force supervision, possibly derived from various electronic structure methods, making the physical reliability of Hessians from pre-trained foundation models questionable.
	* Notably, the authors mention that some student models outperform foundation models in specialized chemical spaces even before distillation, suggesting that foundation models may not fully converge in certain cases. This raises questions about the significance and reliability of using Hessians from foundation models as distillation targets.
	* Accurate Hessians are crucial for tasks like geometry optimization (as referenced in Figure 1). It remains unclear how potentially inaccurate Hessians from foundation models could affect the student model's performance in such applications.
* It is uncertain whether the proposed method can be extended to MLFF architectures designed with energy-conserving forces or high-order equivariance, which are often crucial factors for stable and transferable ML force fields. Discussing the impact of these inductive biases on the Hessian-driven KD approach would strengthen the work.

**Questions:**

See “Weaknesses” for detailed questions and suggestions.

Potential typos:
* Line 420: “disilling” should be “distilling.”

**Details Of Ethics Concerns:**

No ethics concerns.

---

> ### Author Response · Authors · 2024-11-20
> **Response**
>
> **Question: Rationale/insights on using the Hessian for distillation.**
>
> Please refer to Section 3.1: “Background on Energy Hessians”, where we discuss the physical interpretation and motivation of using energy Hessians. In short, the Hessian captures essential information about the curvature of the potential energy surface and vibrational modes. Hessians are also directly used in many geometry optimization/structure relaxation algorithms based on Quasi-Newton dynamics. Also note our reference to Sobolev training - which formalizes several favorable theoretical properties of learning from function derivatives, including better sample complexity and reduced overfitting - in the final paragraph (“Learning from Function Derivatives”) of Section 2.
>
> **Question: Some student models outperform foundation models before distillation, raising concerns about reliability of Hessians from foundation models.**
>
> We agree that student models outperforming the foundation models even before distillation is indicative of a suboptimal/unconverged foundation model. However, we point out that Hessian distillation still leads to improvements in these cases (e.g. the GemNet-dT student models on the MPTraj splits in Table 2), indicating that there is some useful regularization signal from the Hessian term even if the foundation model has higher errors than the undistilled model.
>
> On a broader note, we posit that continued model and data scaling will likely lead to improved  MLFF foundation models in the coming years. If the lessons from CV and NLP hold, we should expect the multi-task performance of these FMs to correspondingly improve, which would in turn lead to better student models after distillation. In fact, we have already demonstrated a glimpse of this phenomenon with our JMP results in Table 3: distilling with Hessians from JMP-Large (220M parameters) generally leads to better student performance than distilling from JMP-Small (39.9M parameters).
>
> **Question: Impact of potentially inaccurate Hessians from foundation models on geometry optimization.**
>
> We have added a geometry optimization experiment in the Appendix Section A.13.  Using our undistilled and distilled GemNet-dT student models, as well as the MACE-OFF FM, we perform geometry optimization on 100 structures from the test set of the Monomers subset of SPICE. We evaluate the energy and forces of each final, optimized structure using DFT at the same level of theory used in the SPICE dataset. We find that on average, our distilled GemNet-dT model converges to structures with lower energy and per-atom force norms than its undistilled counterpart. Taken together with our results on NVE and NVT MD simulations (Figures 2 and 4 respectively), this shows the promise of using distilled MLFFs for several downstream applications, particularly as foundation models continue to get better in the future with model and data scale.
>
> **Question: Extension to models with energy-conserving forces or higher-order equivariance.**
>
> Please refer to Section 4.3 (“Distilling JMP on MD22”) of the updated paper. We have included a series of new experiments where we use JMP, a large foundation model which uses non-conservative forces, as the teacher model, and GemNet-T and eSCN, which use conservative forces and higher order (l=2) equivariance respectively, as student models. We find that the Hessian distillation approach works well in this setting, leading to strong improvements in Force MAE over undistilled baselines (Table 3). We also find that the JMP-L foundation model does not conserve energy well in NVE MD simulations, while our distilled GemNet-T models do so by construction (Figure 3). This highlights the potential usefulness of distilling into student models with inductive biases suited to the downstream task at hand, even if the foundation model lacks these inductive biases.
>
> **Typo: “disilling” should be “distilling.”**
>
> Fixed. Thanks for catching this.

---

> > ### Comment · Reviewer_8NSg · 2024-11-29
> > **Response**
> >
> > The authors' rebuttal has addressed most of my concerns. I have decided to raise my score from 5 to 6.

---

### Official Review · Reviewer_NWzu · 2024-11-04

**Soundness:** 3
**Presentation:** 3
**Contribution:** 3
**Rating:** 8
**Confidence:** 4

**Summary:**

This work proposes a new method for improving the efficiency of Machine Learning Force Fields (MLFFs) by utilizing a knowledge distillation technique. The method distills knowledge from large, general-purpose foundation MLFF models (FMs) into smaller, faster MLFFs specialized for specific regions of chemical space. This is accomplished by aligning the energy Hessians, which are the second derivatives of the energy with respect to atomic positions, between the teacher FM and the student MLFF. By strategically subsampling rows of the Hessian, the authors significantly reduce the computational cost of the distillation process. The authors demonstrate that their approach can achieve speedups of up to 20 times compared to the original FMs while retaining, and in some cases exceeding, the accuracy of the foundation models.

**Strengths:**

- The distilled MLFFs are up to 20x faster than their foundation model counterparts, enabling more efficient simulations.
- The distilled MLFFs achieve comparable or even better force prediction accuracy than the original FMs and demonstrate improved MD stability results.
- The Hessian distillation method is model architecture agnostic.
- Subsampling hessian rows significantly reduces computational costs without sacrificing performance. Its also interesting that subsampling quality doesn't impact the performance much.

**Weaknesses:**

- Training with hessian distillation increases the computational cost compared to undistilled training.
- An anonymous link to the code is not available.

**Questions:**

- Some models use conservative forces and some don't. Do you have a sense of how much that impacts when you distill Hessians from a non-conservative model instead or if the student model is conservative? Or do you expect that not to impact performance as it seems like hessian quality doesn't?
- Why are the rows sampled so different for GemNet across training set and the same for PaiNN? What would be the general suggestion to set this hyperparameter? Or should everyone be iterating on this for every dataset, model architecture, etc?
- Why don't you compare the results with Kelvinius et al. 2023 on OC20-2M and COLL on which the original work was performed?

---

> ### Author Response · Authors · 2024-11-20
> **Response**
>
> **Question: Computational cost of Hessian distillation**
>
> We agree that our distillation procedure increases the cost relative to undistilled training, but as we mention in our paper, in comparison to the cost of training the original foundation model, this cost is fairly minimal (under 10%). We have also demonstrated in Appendix Section A.9 that it is possible to reduce the computational cost by up to 40% for gradient-based student models by computing the Hessians via finite differences instead of auto-differentiation.
>
> **Question: Impact of conservative vs non-conservative forces in student and teacher models.**
>
> Please refer to Section 4.3 (“Distilling JMP on MD22”) of the updated paper. We have included a series of new experiments where we use JMP, a large foundation model which uses non-conservative forces, as the teacher model, and GemNet-T, which uses conservative forces, as the student model. We find that the Hessian distillation approach works well in this setting, leading to strong improvements in Force MAE over undistilled baselines (Table 3). We also find that the JMP-L foundation model does not conserve energy well in NVE MD simulations, while our distilled GemNet-T models do so by construction (Figure 2). This highlights the potential usefulness of distilling into student models with inductive biases suited to the downstream task at hand, even if the foundation model lacks these inductive biases.
>
> **Question: Sampled rows different across models/training sets.**
>
> We selected this hyperparameter with computational efficiency in mind. GemNet-dT, being significantly slower than PaiNN, experiences a notable increase in training time with larger sampling sizes. Consequently, for larger datasets, we reduced the number of sampled rows  for GemNet, while maintaining a constant sampling size for PaiNN even as dataset sizes increased. To achieve greater consistency, we have now standardized GemNet-dT sampling size by reducing it from 10 to 4 for the Monomers and Iodine datasets, without any loss in performance. For the Solvated Amino Acids dataset, we further reduced the sampling size to 1, as it did not impact the results. Table 10 of the appendix shows our updated number of sampled rows.
>
> **Question: Why aren’t comparisons done with Kelvinius et al. 2023 on OC20-2M and COLL?**
>
> For both of these datasets, we did not find a natural way to create chemical relevant subsets on which to train specialized student MLFFs. Additionally, to our knowledge, there are no publically available, pretrained foundation model MLFFs which include COLL as part of their training data. We emphasize that we have included the n2n method introduced in Kelvinius et al as a baseline for our SPICE and MPtrj experiments, and we show that we clearly outperform it.
>
> **Question: No anonymous code link.**
>
> We plan to release the code publicly before the decision period is over.

---

> > ### Comment · Reviewer_NWzu · 2024-11-21
> > **Response**
> >
> > Thanks for the detailed response and for performing additional experiments! The response clarifies all my concerns with computational cost, inconsistent hyperparameter sampling, and comparisons to prior work. I'd still like to see results on OC20-2M as its the only large and diverse datasets and the community can benefit more from distilled models on this (however, I don't consider this to be a reason to block this work from acceptance). Additional experiments going from a non-conservative teacher model to conservative student model, and acceleration of hessians with finite differences clearly increases the impact of this work. Therefore, I've updated my score to a clear accept!

---

> > > ### Author Response · Authors · 2024-11-21
> > > **Response**
> > >
> > > Thank you for your feedback and for raising your score! We agree that OC20-2M would be an interesting and useful setting in which to demonstrate our method. In initial experiments, we found it difficult to work with this dataset due to the lack of natural chemical subset splits, and the large diversity of chemical formulas present in the dataset. However, we will definitely continue to pursue this in future work.

---

### Author Response · Authors · 2024-11-20
**General Response**

We thank the reviewers for their helpful comments on our work, which introduces a new Hessian distillation approach to produce fast, specialized machine learning force fields (MLFFs) from large, general-purpose foundation models (FMs). We appreciate that the reviewers found the paper to be clear, well-written, and showed strong results. Below, we highlight the major improvements and changes we have made to the paper. We have also uploaded a new version of the paper with changes indicated in blue text.

1. **Distilling from a non-conservative teacher model to a conservative or higher-order equivariant student model.** Several reviewers were interested in how our Hessian distillation approach performs when the student model uses gradient-based/conservative forces, while the teacher model uses direct/non-conservative forces. We have addressed this by adding a Section (4.3) to our paper focusing on JMP [1], a large, non-conservative foundation model, as the teacher model, with conservative GemNet-T models as the students. We achieve strong improvements in force MAE compared to undistilled models on the buckyball catcher and double walled nanotube molecules from MD22. These are molecules which are sufficiently large that running the original JMP foundation model with conservative forces is memory-prohibitive on a standard GPU. Crucially, when we run the distilled models in constant energy (NVE) MD simulations, the JMP model’s energy gradually drifts, while our GemNet-T student model conserves energy by design. Although we found the expense of distilling conservative student models to be manageable in practice, it is possible to accelerate training using finite differences (see point 4 below). We also demonstrate strong results when using eSCN [2], which employs higher-order (l=2) equivariance, as the student model. This clearly suggests that our approach is useful in the relevant setting where the student models have more inductive biases than the teacher FM.

2. **Energy results.** In addition to showing results on Force MAE, we have now added results on Energy MAE for the SPICE dataset (Table 1). We find that our Hessian distillation approach, along with a new loss term involving the gradient of the energy prediction head of non-conservative models (defined in Section 3.4), yields strong improvements on Energy MAE relative to undistilled student MLFFs and our n2n and a2a baselines. We also provide an ablation study in the Appendix (Section A.11) to show that pure Hessian distillation without this additional loss term also leads to improvements in energy MAE.

3. **Geometry optimization**. As another demonstration of the downstream usefulness of our method in realistic use cases, we perform geometry optimization with our undistilled and distilled student models on selected systems from the SPICE dataset, and find that our distilled models converge to structures with lower energies and force norms (evaluated with DFT) than their undistilled counterparts (Section A.13).

4. **Accelerating Hessian computation with finite differences**. We show in Appendix Section A.9 that we can replace auto-differentiation with finite differences to accelerate Hessian computation by approximately 40% for gradient-based student models.

5. **Hyperparameter sweeps/baseline tuning.** We have performed sweeps over selected hyperparameters, including the knowledge distillation weight and loss scheduling, for our Hessian distillation method and the baselines we compare to. This has improved the performance of our baselines, making them in line with results reported in [3] but still considerably inferior to our approach.

[1] Shoghi, Nima, et al. "From molecules to materials: Pre-training large generalizable models for atomic property prediction." International Conference on Learning Representations, 2024.

[2] Passaro, Saro, and C. Lawrence Zitnick. "Reducing SO (3) convolutions to SO (2) for efficient equivariant GNNs." International Conference on Machine Learning. PMLR, 2023.

[3] Ekström Kelvinius, Filip, et al. "Accelerating molecular graph neural networks via knowledge distillation." Advances in Neural Information Processing Systems 36 (2024).

---

### Meta-Review · Area_Chair_zRB8 · 2024-12-19

**Metareview:**

The submission presents a distillation method for machine learning force field (MLFF). The notable feature of the method is that the guidance from the teacher MLFF is provided in the form of the energy Hessian. A few dedicated techniques are introduced for better efficiency, e.g., row subsampling and the Jacobian-vector product formulation. I'm glad to see a work that considers a practical and promising situation of distilling a specialized, light-weighted MLFF from an MLFF foundation model, and noting the value of the energy Hessian. The reviewers also agree on this contribution.

Reviewers raised a few concerns and insufficiencies of the paper:
* The common ask is the effectiveness of the method when the inductive bias does not match: using a non-conservative MLFF teacher models and student model that is conservative. The authors have provided additional results in such cases, which appropriately demonstrate the effectiveness.
* The authors provided more details on hyperparameter choices (in response to Reviewers 8TDr and NWzu) for the consideration to support that the better results do not come from unfair tuning on baselines. This seems reasonably convincing.
* Reviewer 8NSg challenged the reliability of the Hessian of the teacher model. In response, the authors mentioned a related empirical evidence in one case, and refer to scaling law to relief the problem, which seem to have satisfied the reviewer.
* The authors provided energy evaluation results in one case in response to Reviewer yqHS, which are supportive.
* Reviewer yqHS challenged the effectiveness of the method under the spirit of derivative supervision by asking for a comparison with autograd force MLFF model which already has its derivatives supervised. The authors provided supportive results for the proposed Hessian distillation method in one case.
* The authors provided details on additional computational cost in response to Reviewer NWzu, and the results seem acceptable.
* I came up with the question that, while the energy Hessian does hold physical relevance, it is also relevant to provide interpretations on why it is also a good choice for distilling from a teacher MLFF. In the rebuttal to Reviewer 8NSg, they provided additional interpretations in terms of Sobolev training, which sounds a reasonable supplement.

After the rebuttal period, all reviewers increased their scores, and all the scores are positive. This suggests that all the major concerns and insufficiencies are addressed. I hence recommend accepting the paper. To make the paper stronger, I would suggest the followings:
* Regarding the reliability of the Hessian of the teacher model, the authors may consider more discussions since the Hessian of a neural network could still largely depend on a specific architecture: the Hessian of the teacher model is neither trained by data, and certain architectures may restrict the expressiveness of Hessian. The authors may consider discussing what kind of architectures can be considered.
* The authors may be interested in discussing the relation to distillation methods that align the latent representation of the two models.
* Reviewer yqHS asked about the setting for the comparison with teacher force distillation. In the rebuttal, although it is clear that there is no conflict in the force labels, but this setting seems to lose the ground-truth force supervision. The authors may consider further clarifying the setting in the section or table caption, and how it is a fair comparison with Hessian distillation.
* Some issues asked by reviewers are only demonstrated in one case. Hope the authors could provide general explanations on these issues, or provide results in more cases, if possible.

**Additional Comments On Reviewer Discussion:**

Reviewers raised a few concerns and insufficiencies of the paper:
* The common ask is the effectiveness of the method when the inductive bias does not match: using a non-conservative MLFF teacher models and student model that is conservative. In the rebuttal, the authors have provided additional results in such cases, which appropriately demonstrate the effectiveness.
* In the rebuttal, the authors provided more details on hyperparameter choices (in response to Reviewers 8TDr and NWzu) for the consideration to support that the better results do not come from unfair tuning on baselines. This seems reasonably convincing.
* Reviewer 8NSg challenged the reliability of the Hessian of the teacher model. In response, the authors mentioned a related empirical evidence in one case in the rebuttal, and refer to scaling law to relief the problem, which seem to have satisfied the reviewer.
* In the rebuttal, the authors provided energy evaluation results in one case in response to Reviewer yqHS, which are supportive.
* Reviewer yqHS challenged the effectiveness of the method under the spirit of derivative supervision by asking for a comparison with autograd force MLFF model which already has its derivatives supervised. In the rebuttal, the authors provided supportive results for the proposed Hessian distillation method in one case.
* The authors provided details on additional computational cost in response to Reviewer NWzu, and the results seem acceptable.
* I came up with the question that, while the energy Hessian does hold physical relevance, it is also relevant to provide interpretations on why it is also a good choice for distilling from a teacher MLFF. In the rebuttal to Reviewer 8NSg, they provided additional interpretations in terms of Sobolev training, which sounds a reasonable supplement.

After the rebuttal period, all reviewers increased their scores, and all the scores are positive. This suggests that all the major concerns and insufficiencies are addressed. I hence recommend accepting the paper.

---

### Decision · Program_Chairs · 2025-01-22

Accept (Poster)